



# Microbial functional signature in the atmospheric boundary layer

Romie Tignat-Perrier[1,2*], Aurélien Dommergue[1], Alban Thollot[1], Olivier Magand[1], Timothy M. Vogel[2], Catherine Larose[2]

[1]Institut des Géosciences de l'Environnement, Université Grenoble Alpes, CNRS, IRD, Grenoble INP, Grenoble, France
[2]Environmental Microbial Genomics, Laboratoire Ampère, École Centrale de Lyon, Université de Lyon, Écully, France

*Correspondence to:* Romie Tignat-Perrier (romie.tignat-perrier@univ-grenobe-alpes.fr )

**Abstract**

Microorganisms are ubiquitous in the atmosphere and some airborne microbial cells were shown to be particularly resistant to atmospheric physical and chemical conditions (*e.g.,* UV radiation, desiccation, presence of radicals). In addition to surviving, some cultivable microorganisms of airborne origin were shown to be able to grow on atmospheric chemicals in laboratory experiments. Metagenomic investigations have been used to identify specific signatures of microbial functional potential in different ecosystems. We conducted a preliminary comparative metagenomic study on the overall microbial functional potential and specific metabolic and stress-related microbial functions of atmospheric microorganisms in order to determine whether airborne microbial communities possess an atmosphere-specific functional potential signature as compared to other ecosystems (*i.e.* soil, sediment, snow, feces, surface seawater *etc*.). In absence of a specific atmospheric signature, the atmospheric samples collected at nine sites around the world were similar to their underlying ecosystems. In addition, atmospheric samples were characterized by a relatively high proportion of fungi. The higher proportion of sequences annotated as genes involved in stress-related functions (*i.e.* functions related to the response to desiccation, UV radiation, oxidative stress *etc*.) resulted in part from the high concentrations of fungi that might resist and survive atmospheric physical stress better than bacteria.

**Keywords**: atmospheric microorganisms, airborne microbial communities, planetary boundary layer, metagenomic sequencing, comparative metagenomics, selective processes

## 1 Introduction

Microorganisms are ubiquitous in the atmosphere and reach concentrations of up to $10^6$ microbial cells per cubic meter of air (Tignat-Perrier et al., 2019). Due to their important roles in public health and meteorological processes (Ariya et al., 2009; Aylor, 2003; Brown and Hovmøller, 2002; Delort et al., 2010; Griffin, 2007), understanding how airborne microbial communities are distributed over time and space is critical. While the concentration and taxonomic diversity of airborne microbial communities in the planetary boundary layer have recently been described (Els et al., 2019; Innocente et al., 2017; Tignat-Perrier et al., 2019), the functional potential of airborne microbial communities remains unknown. Most studies have focused on laboratory cultivation to identify possible metabolic functions of microbial strains of atmospheric origin, mainly from cloud water (Amato et al., 2007; Ariya et al., 2002; Hill et al., 2007; Vaïtilingom et al., 2010, 2013). Given that cultivatable organisms represent about 1 % of the entire microbial community (Vartoukian et al., 2010), culture-independent techniques and especially metagenomic studies applied to atmospheric microbiology have the potential to provide additional information on the selection and genetic adaptation of airborne





microorganisms. However, to our knowledge, only five metagenomic studies on airborne microbial communities at one or two specific sites per study exist (Aalismail et al., 2019; Amato et al., 2019; Cao et al., 2014; Gusareva et al., 2019; Yooseph et al., 2013). Metagenomic investigations of complex microbial communities in many ecosystems (for example, soil, seawater, lakes, feces, sludge) have provided evidence that microorganism functional signatures reflect the abiotic conditions of their environment, with different relative abundances of specific microbial functional classes (Delmont et al., 2011; Li et al., 2019; Tringe et al., 2005; Xie et al., 2011). This observed correlation of microbial community functional potential and the physical and chemical characteristics of their environments could have resulted from genetic modifications (microbial adaptation) (Brune et al., 2000; Hindré et al., 2012; Rey et al., 2016; Yooseph et al., 2010) and/or physical selection. The latter refers to the death of sensitive cells and the survival of resistant or previously adapted cells. This physical selection can occur when microorganisms are exposed to physiologically adverse conditions.

The presence of a specific microbial functional signature in the atmosphere has not been investigated yet. Microbial strains of airborne origin have been shown to survive and develop under conditions typically found in cloud water (*i.e.* high concentrations of $H_2O_2$, typical cloud carbonaceous sources, UV radiation *etc.*) (Amato et al., 2007; Joly et al., 2015; Vaïtilingom et al., 2013). While atmospheric chemicals might lead to some microbial adaptation, physical and unfavorable conditions of the atmosphere such as UV radiation, low water content and cold temperatures might select which microorganisms can survive in the atmosphere. From the pool of microbial cells being aerosolized from Earth's surfaces, these adverse conditions might act as a filter in selecting cells already resistant to unfavorable physical conditions. Fungal cells and especially fungal spores might be particularly adapted to survive in the atmosphere due to their innate resistance (Huang and Hull, 2017) and might behave differently than bacterial cells. Still, the proportion and nature (*i.e.* fungi versus bacteria) of microbial cells that are resistant to the harsh atmospheric conditions within airborne microbial communities are unknown.

Our objective was to determine whether airborne microorganisms in the planetary boundary layer possess a specific functional signature as compared to other ecosystems since this might indicate that microorganisms with specific functions tend to be more aerosolized and/or undergo a higher survival in this environment. Our previous study showed that airborne microbial taxonomy mainly depends on the underlying ecosystems, indicating that the local environments are the main source of airborne microorganisms (Tignat-Perrier et al., 2019). Still, we do not know if airborne microbial communities result from random or specific aerosolization of the underlying ecosystems' microorganisms. We used a metagenomic approach to compare the differences and similarities of both the overall functional potential and specific microbial functions (metabolic and stress-related functions) between microbial communities from the atmosphere and other ecosystems (soil, sediment, surface seawater, river water, snow, human feces, phyllosphere and hydrothermal vent). We sampled airborne microbial communities at nine different locations around the world during several weeks to get a global-scale view and to capture the between and within-site variability in atmospheric microbial functional potential.

## 2 Material and Methods

### 2.1 Sites and sampling

Air samples were collected at nine sites in 2016 and 2017. Sites were characterized by different latitudes (from the Arctic to the sub-Antarctica; **Fig 1**), elevations from sea level (from 59 m to 5230 m; **Fig 1**) and environment type (from marine for Amsterdam-Island or AMS, to coastal for Cape Point or CAP, polar for Station Nord or STN and terrestrial for Grenoble or GRE, Chacaltaya or CHC, puy de Dôme or PDD, Pic-du-Midi or PDM, Storm-Peak or STP and Namco or NAM - **Table S1**). The number of samples collected per site varied from seven to





sixteen (**Table S1**). We collected particulate matter smaller than 10 μm (PM10) on quartz fiber
filters (5.9'' round filter and 8'' × 10'' rectangular types) using high volume air samplers
(TISCH, DIGITEL, home-made) installed on roof tops or terraces (roughly 10 m above ground
level). To avoid contamination, quartz fiber filters as well as all the material in contact with the
filters (*i.e.* filter holders, aluminium foils and plastic bags in which the filters were transported)
were sterilized using strong heating (500 °C for 8 h) and UV radiation, respectively as detailed
in Dommergue et al., 2019. The collection time per sample lasted one week, and the collected
volumes ranged from 2000 $m^3$ to 10000 $m^3$ after standardization using SATP standards
(Standard Ambient Pressure and Temperature). Detailed sampling protocols including negative
control filters are presented in Dommergue et al. 2019. MODIS (Moderate resolution imaging
spectroradiometer) land cover approach (5' x 5' resolution) (Friedl et al., 2002; Shannan et al.,
2014) was used to quantify landscapes in the 50 km diameter area of our nine sampling sites
(**Fig S1**).

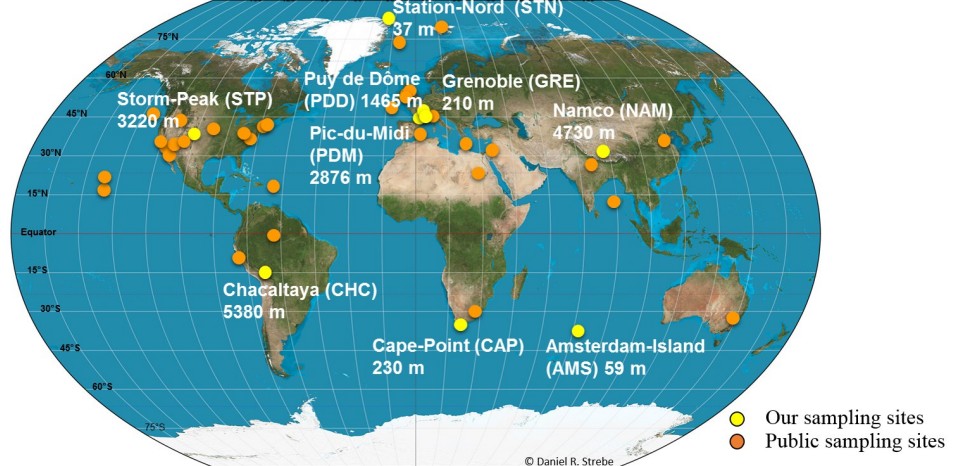

**Fig 1**. **Sample collection locations**. Map showing the geographical location and elevation from
sea level of our nine sampling sites (in yellow), and the geographical position of whose public
metagenomes come from (in orange). Abbreviations of our nine sampling sites are indicated in
brackets.

**2.2 Molecular biology analyses**
**2.2.1 DNA extraction**
DNA was extracted from three circular pieces (punches) from the quartz fiber filters (diameter
of one punch: 38 mm) using the DNeasy PowerWater kit with some modifications as detailed
in Dommergue et al., 2019. During cell lysis, the PowerBead tube containing the three punches
and the pre-heated lysis solution were heated at 65 °C during one hour after a 10-min vortex
treatment at maximum speed. We then separated the filter debris from the lysate by
centrifugation at 1000 rcf for 4 min. From this step on, we followed the DNeasy PowerWater
protocol. DNA concentration eluted in 100 μL of buffer was measured using the High Sensitive
Qubit Fluorometric Quantification (Thermo Fisher Scientific). DNA was stored at -20 °C.

**2.2.2 Real-Time qPCR analyses**





***16S rRNA gene qPCR.*** The bacterial cell concentration was approximated by the number of 16S rRNA gene copies per cubic meter of air. The V3 region of the 16S rRNA gene was amplified using the SensiFast SYBR No-Rox kit (Bioline) and the following primers sequences: Eub 338f 5'-ACTCCTACGGGAGGCAGCAG-3' as the forward primer and Eub 518r 5'-ATTACCGCGGCTGCTGG-3' as the reverse primer (Fierer et al., 2005) on a Rotorgene 3000 machine (Qiagen). The reaction mixture of 20 µL contained 10 µL of SYBR master mix, 2 µL of DNA and RNAse-free water to complete the final 20 µL volume. The qPCR 2-step program consisted of an initial step at 95 °C for 2 min for enzyme activation, then 35 cycles of 5 s at 95 °C and 20 s at 60 °C hybridization and elongation. A final step was added to obtain a denaturation from 55 °C to 95 °C with increments of 1 °C s$^{-1}$. The amplicon length was around 200 bp. PCR products obtained from DNA from a pure culture of *Escherichia coli* were cloned in a plasmid (pCR™2.1-TOPO® vector, Invitrogen) and used as standard after quantification with the Broad-Range Qubit Fluorometric Quantification (Thermo Fisher Scientific).

***18S rRNA gene qPCR.*** The fungal cell concentration was estimated by the number of 18S rRNA gene copies per cubic meter of air. The region located at the end of the SSU 18S rRNA gene, near the ITS 1 region, was quantified using the SensiFast SYBR No-Rox kit (Bioline) and the following primers sequences: FR1 5'-AICCATTCAATCGGTAIT-3' as the forward primer and FF390 5'-CGATAACGAACGAGACCT-3' as the reverse primer (Chemidlin Prévost-Bouré et al., 2011) on a Rotorgene 3000 machine (Qiagen). The reaction mixture of 20 µL contained 10 µL of SYBR master mix, 2 µL of DNA and RNAse-free water to complete the final 20 µL volume. The qPCR 2-steps program consisted of an initial step at 95 °C for 5 min for enzyme activation, then 35 cycles of 15 s at 95 °C and 30 s at 60 °C hybridization and elongation. A final step was added to obtain a denaturation from 55 °C to 95 °C with increments of 1 °C s$^{-1}$. The amplicon length was around 390 bp. PCR products obtained from DNA from a soil sample were cloned in a plasmid (pCR™2.1-TOPO® vector, Invitrogen) and used as standard after quantification with the Broad-Range Qubit Fluorometric Quantification (Thermo Fisher Scientific).

### 2.2.3 MiSeq Illumina metagenomic sequencing

***Metagenomic library preparation.*** Metagenomic libraries were prepared from 1 ng of DNA using the Nextera XT Library Prep Kit and indexes following the protocol in Illumina's "Nextera XT DNA Library Prep Kit" reference guide with some modifications for samples with DNA concentrations below 1 ng as follows. The tagmented DNA was amplified over 13 PCR cycles instead of 12 PCR cycles, and the libraries (after indexing) were resuspended in 30 µL of RBS buffer instead of 52.5 µL. Metagenomic sequencing was performed using the MiSeq and V2 technology of Illumina with 2 x 250 cycles. At the end of the sequencing, the adapter sequences were removed by internal Illumina software.

***Reads quality filtering.*** Reads 1 and reads 2 per sample were not paired but merged in a common file before filtering them based on read quality using the tool FASTX-Toolkit (http://hannonlab.cshl.edu/fastx_toolkit/) using a minimum read quality of Q20, minimum read length of 120 bp and one maximum number of N per read. Samples with less than 6000 filtered sequences were removed from the dataset.

### 2.2.4 Downloading of public metagenomes

Public metagenomes were downloaded from the MGRAST and SRA (NCBI) databases as quality filtered read-containing fasta files and raw read containing fastq files, respectively. The fastq files containing raw reads underwent the same quality filtering as our metagenomes (as discussed above). The list of the metagenomes, type of ecosystem, number of sequences and sequencing technology (*i.e.* MiSeq, HiSeq or 454) are summarized in **Table S2**. The sampling sites are positioned on the map in **Fig 1**.





183

## 2.3 Data analyses

All graphical and multivariate statistical analyses were carried out using the vegan (Oksanen et al., 2019), ggplot2 (Hadley and Winston, 2019) and reshape2 (Wickham, 2017) packages in the R environment (version 3.5.1).

### 2.3.1 Annotation of the metagenomic reads

Firstly, to access the overall functional potential of each sample, the filtered sequences per sample were functionally annotated using Diamond, then the gene-annotated sequences were grouped in the different SEED functional classes (around 7000 functional classes, referred simply to as functions) using MEGAN version 6 (Huson et al., 2009). Functional classes that were present ≤ 2 times in a sample were removed of this sample. In parallel, the Kraken software (Wood and Salzberg, 2014) was used to retrieve the bacterial and fungal sequences separately from the filtered sequences using the Kraken bacterial database and FindFungi (Donovan et al., 2018) fungal database (both databases included complete genomes), respectively (and using two different runs of Kraken). Separately, both the bacterial and fungal sequences were also functionally annotated using Diamond and MEGAN version 6 (number of sequences functionally annotated in **Table S3**).

Secondly, for specific metabolic and stress-related functions, we annotated the sequences using eggNOG-Mapper version 1 (Diamond option), then examined specific GO (Gene Ontology) terms chosen based on their importance for microbial resistance to atmospheric-like conditions. The different GO terms used were the following: GO:0042744 (hydrogen peroxide catabolic activity), GO:0015049 (methane monooxygenase activity) as specific metabolic functions and GO:0043934 (sporulation), GO:0009650 (response to UV), GO:0034599 (cell response to oxidative stress), GO:0009269 (response to desiccation) as stress-related functions. The number of hits of each GO term was normalized per 10000 annotated sequences and calculated from all sequences, bacterial sequences and fungal sequences for each sample. The number of sequences annotated by eggNOG-Mapper (Huerta-Cepas et al., 2017) was also evaluated (**Table S3**). The putative concentration of a specific function or functional class in the samples is determined as the concentration of sequences annotated as one of the functional proteins associated to this function (or functional class).

### 2.3.2 Statistical analyses

Observed functional richness and evenness were calculated per sample after rarefaction on all sequences (rarefaction at 2000 sequences), bacterial sequences (rarefaction at 500 sequences) and fungal sequences (rarefaction at 500 sequences). The distribution of the samples was analyzed based on the SEED functional classes (using all sequences). PCoA and hierarchical clustering analysis (average method) were carried out on the Bray-Curtis dissimilarity matrix based on the relative abundances of the different SEED functional classes. SIMPER analyses were used to identify the functions responsible for the clustering of samples in groups. Because of the non-normality of the data, Kruskal-Wallis analyses (non-parametric version of ANOVA) and Dunn's post-hoc tests were used to test the difference between the percentage of fungal sequences as well as the number of hits of each Gene Ontology term (normalized per 10000 annotated sequences) among the different sites and the different ecosystems.

## 3 Results

### 3.1 Percentage of fungal sequences

The percentage of sequences annotated as belonging to fungal genomes (or fungal sequences, as opposed to bacterial sequences) was on average higher in air samples compared to soil ($P<10^{-5}$), snow ($P=10^{-3}$), seawater ($P=0.03$) and sediment samples ($P=10^{-3}$; **Fig 2** and **Table S4**).


Among the air samples, NAM (19%), STN (24%) and CHC (27%) showed the lowest
percentages of fungal sequences on average while STP (88%), GRE (79%), AMS (71%) and
PDD (62%) showed the highest percentages. For the ecosystems that were only represented by
one sample, and therefore, were not evaluated by the Kruskal-Wallis test, we observed average
percentages of fungal reads of 3% in feces, 9% in hydrothermal vents, 19% in river water
samples and 37% in the phyllosphere. Some samples from soil, sediments and seawater such as
French agricultural soil (61%), Peru sediments (53%) and Celtic seawater (53%) had relatively
high percentages of fungal sequences while other samples had less than 50%. The number of
fungal and bacterial cells was also estimated using 16S rRNA and 18S rRNA gene copy
numbers per cubic meter of air, respectively. qPCR results on air samples are available in
Tignat-Perrier et al., 2019. Air samples had ratios between bacterial cell and fungal cell
concentrations from around 4.5 times up to 160 times lower than soil samples (**Table S4**).

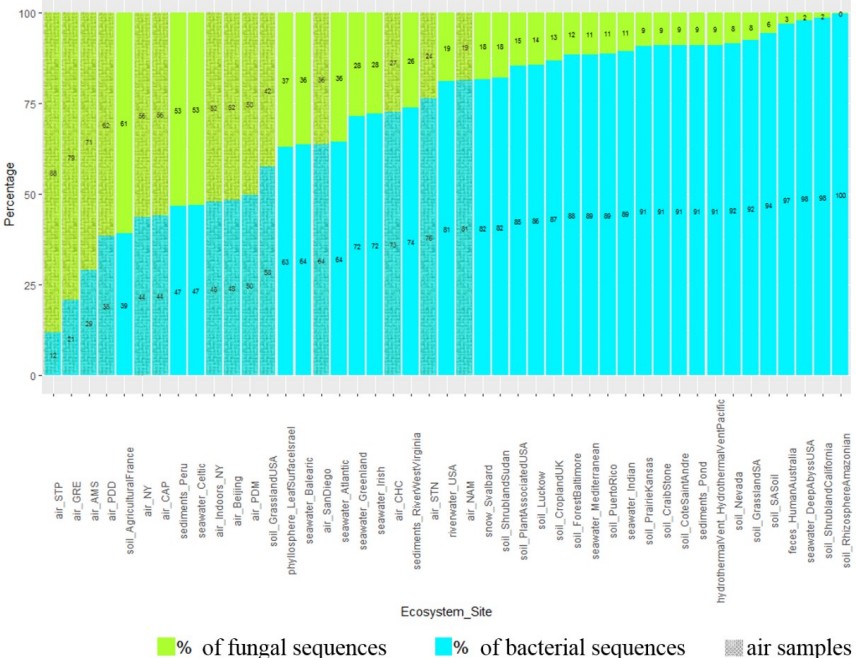


**Fig 2**. **Percentage of fungal and bacterial sequences in the metagenomes.** The percentages
are established as the number of sequences annotated as belonging to fungal and bacterial
genomes over the sum of bacterial and fungal sequences in the metagenomes. The mean was
calculated for the sampling sites including several metagenomes. Air sites (*i.e.* our 9 sites + 5
sites where public air metagenomes come from) are distinguished by grey hatching lines.

**3.2 Airborne microbial functional profiles**
The fifty most abundant SEED functional classes represented in atmospheric samples are listed
in **Table S5**. The 5-FCL-like protein, the long chain fatty acid CoA ligase and the TonB-
dependent receptor were the top three functions based on number of annotated reads observed
when including all the sequences (**Table S5**). The atmospheric microbial functional profiles





based on the SEED functions were compared between samples from the different weeks of
sampling and between different locations. The profiles were graphed using PCo multivariate
analysis to visualize differences and similarities. The different samples (sampled during
sequential weeks) from the same site did not cluster tightly together on the PCo multivariate
analysis. In order to incorporate weekly variation when comparing sites, we used the microbial
functional profile averaged per site in the subsequent multivariate analyses done with the data
from other ecosystems (**Fig 3**). The PCo multivariate analysis showed that terrestrial
atmospheric sites (GRE, NAM, STP, PDD, PDM, CHC, New York) grouped with the soil,
sediment and snow samples while the marine and coastal atmospheric sites (AMS, CAP, San
Diego) were situated between the datasets from soil, seawater and river water (**Fig 3**). The polar
site STN did not group with the other sites. When considering only the bacterial sequences (*i.e.*,
excluding the fungal sequences), the distribution of the terrestrial atmospheric sites did not
change, while the marine Amsterdam-Island, coastal Cape Point and polar Station Nord
atmospheric sites were further from the seawater and river water datasets than when the fungal
sequences were included (**Fig S2**). The distribution of the different datasets underwent further
changes when considering only the fungal sequences. We observed an absence of a clear
separation between soil and seawater since they (for the majority) grouped closely together, and
terrestrial atmospheric datasets did not group with the other non-atmospheric datasets from soil,
sediment and snow (**Fig S2**).

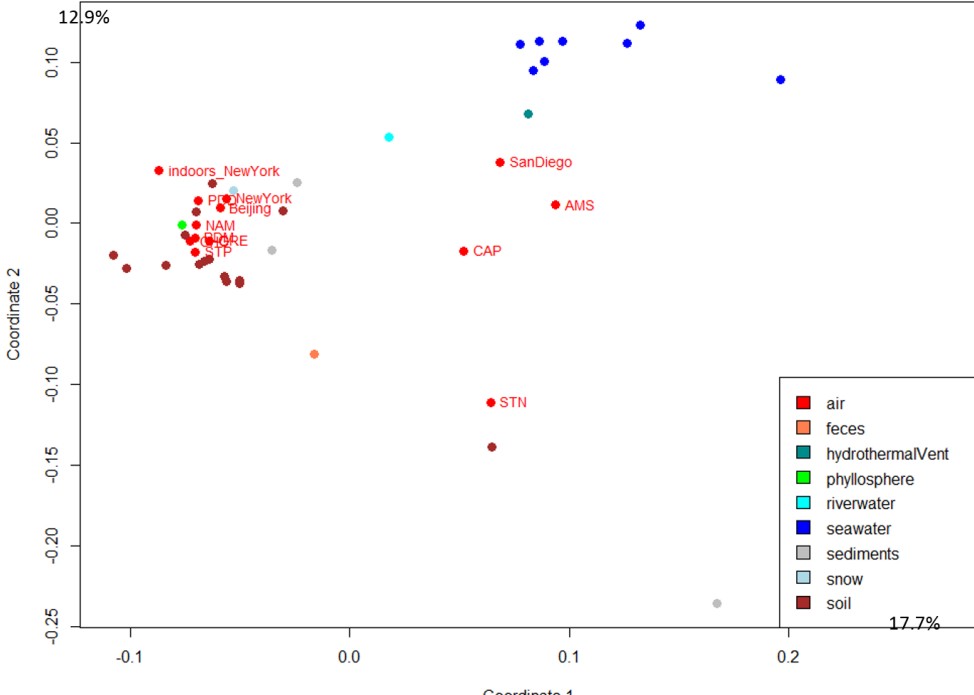

**Fig 3. Distribution of the samples based on the microbial functional profile.** The PCo
analysis of the Bray-Curtis dissimilarity matrix is based on the functional potential structure of
each site. For the site including several metagenomes, the average profile was calculated. Colors
indicate the ecosystems in which the sites belong to.



### 3.3 Airborne microbial functional richness and evenness

Functional richness and evenness were evaluated using the relative abundance of sequences in the different SEED categories. The average richness in SEED functional classes (or functions) in the PBL was lower than the average functional class richness in soil, surface seawater, hydrothermal vents, river water, phyllosphere and feces ($P<0.05$) (**Table S3**). Among the different atmospheric samples, the functional class richness was highest in Beijing (4060 +/- 112 functional classes) and New York indoor air samples (3302 +/- 299 functional classes) ($P<0.05$), and lowest in Station Nord (956 +/- 547 functional classes). When looking at the bacteria-annotated sequences, almost the same trend was observed, *i.e.* the functional class richness in air was lower than in soil, hydrothermal vents, river water, phyllosphere and feces, and not different from the other ecosystems ($P<0.05$ and $>0.05$, respectively) (**Table S3**). The functional class richness was higher in Beijing (2835 +/- 59 functional classes) and New York indoor air samples (2183 +/- 387 functional classes) compared to the other air samples whose values ranged between 270 +/- 197 functional classes in Amsterdam-Island and 1142 +/- 461 functional classes in Chacaltaya. For fungal sequences, the functional class richness in the atmosphere was lower than the functional class richness in soil, surface seawater, feces, hydrothermal vents, river water and phyllosphere ($P<0.05$) (**Table S3**). Within air samples, the functional class richness based on fungal sequences was higher in Beijing (1129 +/- 92 functional classes) and New York indoor air samples (687 +/- 206 functional classes) than in the other air sites ($P<10^{-5}$) whose values ranged from 66 +/- 58 functional classes in Amsterdam-Island and 392 +/- 131 functional classes in Storm Peak (**Table S3**). The functional class evenness in air was on average higher than in soil ($P=0.03$), and not different to the functional class evenness observed in the other ecosystems (sediment, seawater, snow). When looking at the bacterial and fungal sequences separately, the functional class evenness in air was on average higher than in soil, feces, phyllosphere and riverwater ($P<0.05$) (**Table S3**).

### 3.4 Concentration of specific microbial functions that might have a role under atmospheric conditions

Two metabolic functions associated with abundant atmospheric chemicals ($H_2O_2$ and $CH_4$) were examined, hydrogen catabolism and methane monooxygenase activity. The concentration of sequences annotated as hydrogen peroxide catabolic related functional proteins per 10000 sequences varied between air sites ($P=2\times10^{-5}$) with highest values for Amsterdam-Island (27 +/- 1) and Grenoble (27 +/- 1) (**Fig S3**). It was on average higher in air compared to soil ($P=10^{-4}$) and surface seawater ($P=10^{-4}$). The French agricultural soil showed the highest relative abundance (133 +/- 4). When considering the fungal and bacterial sequences separately, this concentration was not different between air and the other ecosystems ($P>0.05$) (**Fig S3**). The number of sequences annotated as methane monooxygenase-related functional proteins per 10000 sequences was only detectable when considering all the sequences (*i.e.* bacterial and fungal sequences). The number of sequences annotated as methane monooxygenase-related functional proteins did not vary between air sites ($P>0.05$) while we observed a high variability between sampling periods within sites, but on average it was not different from the ecosystems ($P>0.05$).

Different stress response functions (sporulation, UV response, oxidative stress cell response, desiccation response, chromosome plasmid partitioning protein ParA and lipoate synthase) were examined. The concentration of sequences annotated as sporulation-related functional proteins per 10000 annotated sequences largely varied between air sites ($P=2\times10^{-9}$), with the lowest values observed for Station Nord (7 +/- 9), San Diego (9 +/- 6), Namco (17 +/- 15) and Chacaltaya (26 +/-13), and the highest values observed for Storm Peak (120 +/- 18), Beijing (126 +/- 22), Grenoble (131 +/- 21) and New York (141 +/- 98) (**Fig 4**). It was on average higher





in air compared to soil ($P<10^{-5}$), sediments ($P<10^{-5}$) and surface seawater ($P=4\times10^{-4}$) although
the Celtic seawater sample presented a very high concentration (127). Snow showed a relatively
high average concentration (*i.e.* 36) which was not different from air concentration ($P>0.05$).
For the ecosystems including one value (*i.e.* one sample, so not integrated in the Kruskal-Wallis
tests), feces showed a relatively high concentration of sequences annotated as sporulation-
related functional proteins (*i.e.* 41) while hydrothermal vent, phyllosphere and river water
showed relatively low concentrations compared to air (<10). When considering the fungal
sequences separately from the bacterial sequences, the same trend was observed, *i.e.* the
concentration of sequences annotated as sporulation-related functional proteins in air was on
average higher compared to soil ($P<10^{-5}$), sediments ($P<10^{-5}$), surface seawater ($P=7\times10^{-4}$) as
well as phyllosphere, hydrothermal vent and river water. The concentration was relatively high
in the Celtic seawater (186) and the snow samples (163 +/- 47). We also observed a large
variability within air sites ($P=3\times10^{-5}$). When considering the bacterial sequences only, this
concentration in air was on average higher compared to soil ($P=0.02$), sediments ($P=4\times10^{-3}$)
and snow ($P=0.01$), and showed a smaller variability between air sites. Two samples, the
phyllosphere (*i.e.* 35) and the shrubland soil from Sudan (*i.e.* 32) showed high numbers of
sequences annotated as sporulation-related functional proteins per 10000 annotated sequences
**(Fig 4)**.



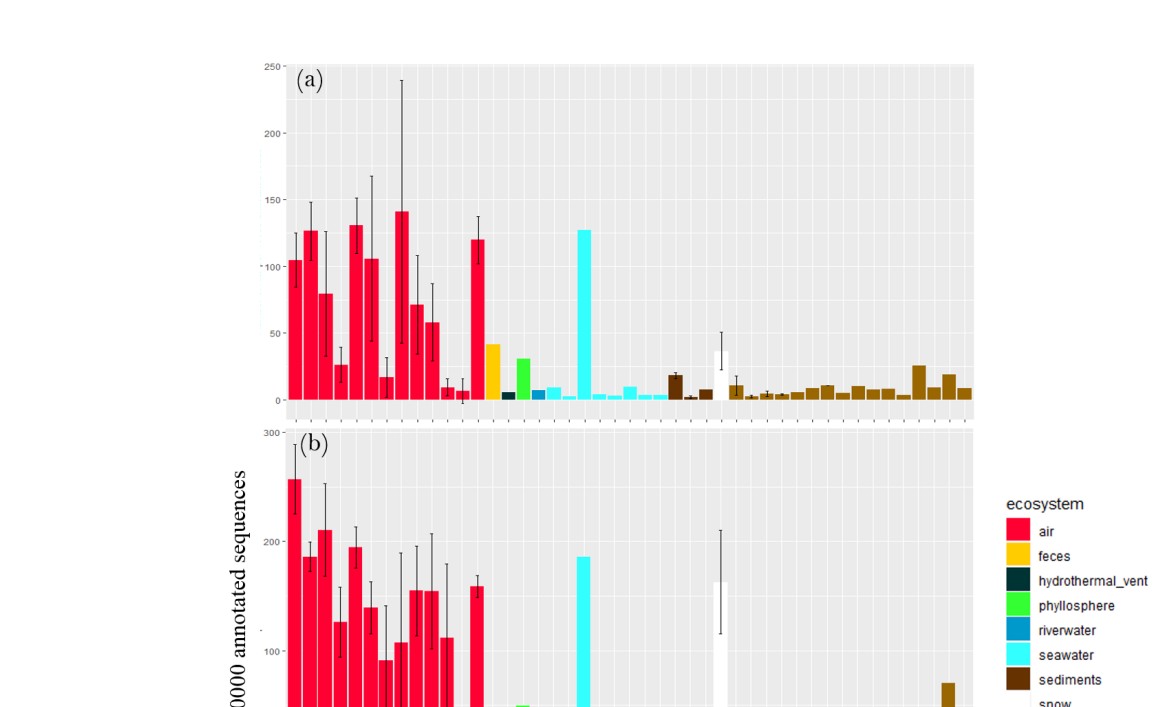

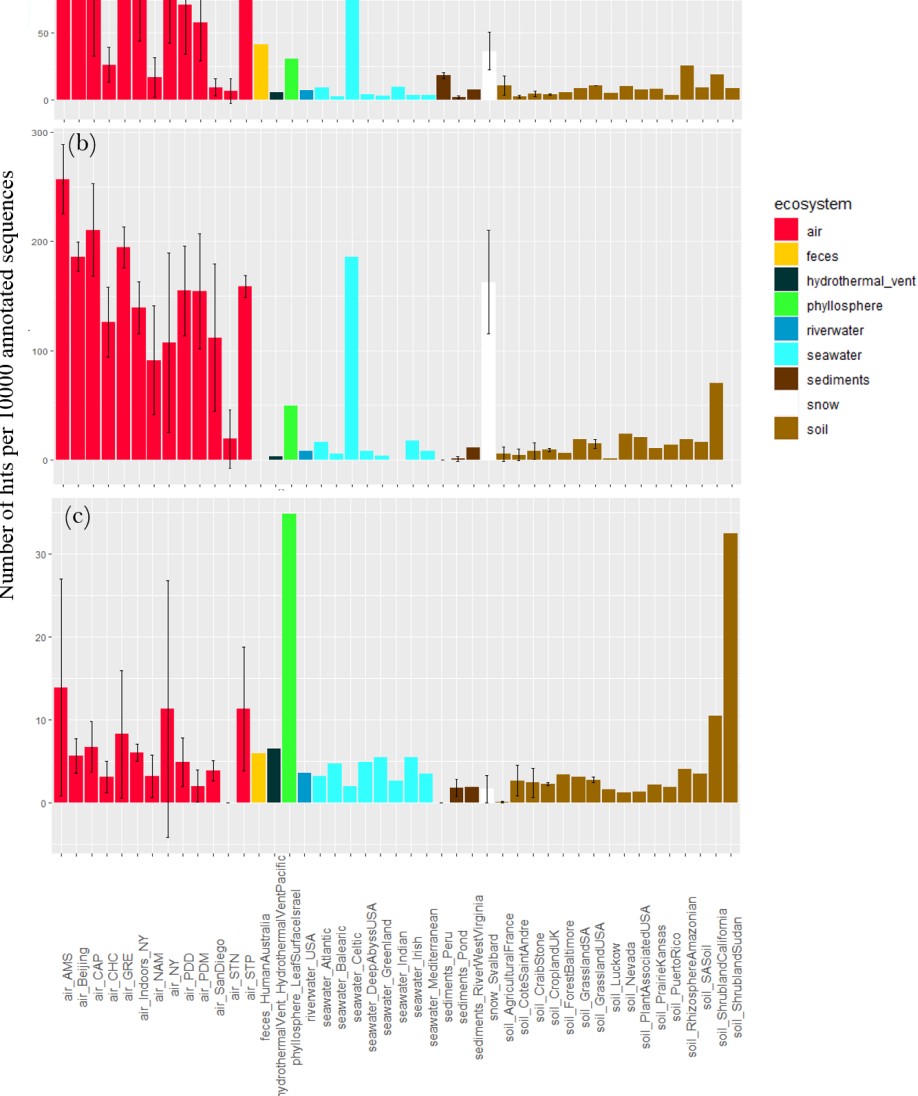


**Fig 4. Proportion of sequences annotated as sporulation related functional proteins in the metagenomes**. Average number of sequences annotated as proteins implicated in sporulation per 10000 annotated sequences from (**a**) all sequences, (**b**) fungal sequences and (**c**) bacterial



sequences per site. Colors indicate the ecosystems in which the sites belong to. For the sites
including several metagenomes, the standard deviation was added.
The concentration of sequences annotated as UV response related functional proteins per 10000
annotated sequences varied between air sites ($P=10^{-5}$), with values ranging from 16 +/- 2 in
Namco and 19 +/- 4 in STN to 29 +/- 3 in Storm Peak and 36 +/- 6 in Amsterdam-Island (**Fig
S4**). The concentration was on average higher in air compared to sediments ($P<10^{-5}$), soil
($P<10^{-5}$) and comparable to snow and surface seawater ($P>0.05$). The other ecosystems showed
lower ratios (feces, phyllosphere) or comparable concentrations (hydrothermal vent, river
water) compared to air. Within the soil samples, the French agricultural soil samples showed a
high average concentration (56 +/- 8), which increased the average ratio observed in soil
samples. When considering fungal sequences separately, the concentration of sequences
annotated as UV response related functional proteins was higher in air compared to soil
($P=9\times10^{-4}$), and comparable to the other ecosystems ($P>0.05$). When considering the bacterial
sequences only, this concentration in air was on average higher compared to seawater ($P=3\times10^{-3}$)
and sediments ($P=6\times10^{-3}$).
The concentration of sequences annotated as oxidative stress cell response related functional
proteins per 10000 annotated sequences varied largely between air sites ($P=5\times10^{-7}$), with the
lowest values observed for Station Nord (23 +/- 5), San Diego (11 +/- 3) and Namco (28 +/-
10), and the highest values observed for Storm Peak (105 +/- 16), Amsterdam-Island (108 +/-
16) and Grenoble (119 +/- 19) (**Fig 5**). The concentration was on average higher in air compared
to soil ($P<10^{-5}$), sediments ($P<10^{-5}$) and surface seawater ($P=2\times10^{-3}$). Snow showed a relatively
high average value (46 +/- 11), not different from air ($P>0.05$). The other ecosystems (feces,
river water, hydrothermal vent, phyllosphere) showed lower ratios compared to air. When
considering fungal sequences separately, the concentration of sequences annotated as oxidative
stress related functional proteins per 10000 sequences was on average higher in air compared
to soil ($P<10^{-5}$), sediments ($P<10^{-5}$) and surface seawater ($P=10^{-3}$). Feces showed a very high
average value (2237). When considering bacterial sequences separately, this concentration was
not different between air and the other ecosystems ($P>0.05$). When considering both fungal and
bacterial sequences separately, the variability in the concentration of sequences annotated as
oxidative stress cell response related functional proteins between air sites diminished and their
difference was not detected anymore ($P>0.05$).









**Fig 5**. **Proportion of sequences as oxidative stress cell response related functional proteins in the metagenomes**. Average number of sequences annotated as proteins implicated in oxidative stress cell response per 10000 annotated sequences from (**a**) all sequences, (**b**) fungal sequences and (**c**) bacterial sequences per site. Colors indicate the ecosystems in which the sites belong to. For the sites including several metagenomes, the standard deviation was added.

The concentration of sequences annotated as desiccation response related functional proteins per 10000 sequences varied between air sites ($P=2\times10^{-5}$), with the highest values in Grenoble (4 +/- 1), Storm Peak (4 +/- 1) and Amsterdam-Island (3 +/- 3), and the lowest values in Station Nord (0.5 +/- 1) and San Diego (0.1 +/- 0.1) (**Fig S4**). It was on average higher in air compared to the other ecosystems ($P=4\times10^{-9}$). Still Svalbard snow and French agricultural soil showed high values (2 +/- 1 and 3 +/- 1, respectively) (**Fig S4**). When considering fungal sequences only, the concentration in air was higher compared to soil ($P>10^{-5}$), sediments ($P>10^{-5}$) and surface seawater ($P=10^{-3}$). No difference between the ecosystems was observed when considering bacterial sequences separately ($P=0.62$).

Two proteins (lipoate synthase and chromosome plasmid partitioning protein ParA) related to stress response showed high relative concentrations in bacterial sequences of a few air samples compared to the other ecosystems (**Fig S3**), although the number of sequences related to these proteins was on average not higher in the atmosphere than other ecosystems ($P>0.05$).

**4 Discussion**

Metagenomic investigations of different ecosystems revealed a specific functional potential signature of their associated microbial communities (Delmont et al., 2011; Tringe et al., 2005). These specific signatures are thought to result from microbial adaptation and/or physical selection to the environmental abiotic conditions (Hindré et al., 2012; Li et al., 2019; Rey et al., 2016) and are a reflection of the high relative abundances of genes coding for specific functions essential for microorganisms to survive and develop in these environments. For example, microbial metagenomes of human feces were characterized by high relative abundances of sequences annotated as beta-glucosidases that are associated with high intestinal concentrations of complex glycosides; and microbial metagenomes of oceans were enriched in sequences annotated as enzymes catalyzing DMSP (dimethylsulfoniopropionate), that is an organosulfur compound produced by phytoplankton (Delmont et al., 2011). Our results showed a clear separation between surface seawater, river water, human feces and almost all the soil samples (which grouped with the sediment and snow samples at the scale used here) on the PCo analysis based on the microbial functional potential (**Fig 3**). For air microbiomes, the PCo analyses showed that the individual air samples did not group for each site and that they did not form a cluster separated from the other ecosystems based on the overall microbial functional potential averaged per site (**Fig 3**). Air samples seemed to group with their underlying ecosystems. While terrestrial air samples (GRE, NAM, CHC, STP, PDD, PDM) grouped with snow, soil and sediment samples, the marine (Amsterdam-Island), coastal (Cape Point) and arctic (Station Nord) air samples were closer to surface seawater and river water samples. Airborne microbial functional potential (and especially metabolic functional potential as SEED functional classes included mainly metabolic functions and few stress response related functions) might be dependent on the ecosystems from which microorganisms are aerosolized. Moreover, it seems that bacterial sequences are mainly responsible for the distribution of the samples on the PCo analysis (as observed when comparing the PCoA to that carried out with the fungal sequences only) although they were in smaller numbers compared to fungal sequences for many of the air samples (*i.e.* STP, GRE, AMS, PDD, CAP, Beijing *etc.*). The low statistical weight of fungal sequences relative to the overall sequences might be related to their low richness in terms of



functional genes that might have resulted in the spreading of the samples on the PCoA based
on the fungal sequences (**Table S3**).
Metagenomes extracted from atmospheric samples taken around the planet were characterized
by a relatively high percentage of fungal sequences as compared to other ecosystems even
though bacterial sequences still dominated. This percentage varied across the different sites
with a higher percentage at terrestrial sites whose surrounding landscapes were vegetated like
Grenoble (GRE), puy de Dôme (PDD) and Pic-du-midi (PDM) (surrounding landscapes in **Fig
S1**). This percentage was also relatively high at the marine site Amsterdam-Island (AMS),
where fungi might come from the ocean and/or the vegetated surfaces of the small island. A
high percentage of fungal sequences was also reported for air samples from Beijing, New York
and San Diego and validates our DNA extraction method set-up specifically for quartz fiber
filter (Dommergue et al., 2019). Similarly, the sequencing technology (Illumina MiSeq) could
not have been responsible for the larger percentage of fungal sequences observed in our datasets
as the Beijing and New York/San Diego air sample datasets originated from Illumina HiSeq
and 454 sequencing technology, respectively. qPCR results on the 16S rRNA gene (bacterial
cell concentration estimation) and on the 18S rRNA gene (fungal cell concentration estimation)
on our air samples in comparison to soil samples (Côte Saint André, France) showed that the
ratio between fungal and bacterial cell number was much higher (from 4.5 to 160 times higher
for the most vegetated site Grenoble) in air than in soil (**Table S4**). The ratio between fungal
and bacterial cell number might be higher in the planetary boundary layer (PBL) than in other
environments like soil (Malik et al., 2016), and thus, would explain the relatively higher
percentage of fungal sequences observed in air metagenomes. High throughput sequencing
allows the sequencing of a small part of the metagenomic DNA (with large fungal genomes
likely to be sequenced first) and might explain why the values of the bacteria and fungi
abundance ratio obtained by qPCR does not match those obtained by the metagenomic
sequencing approach. Our study is a preliminary metagenomic investigation of the air
environment with a limited number of sequences per sample, and further studies are needed to
confirm our results.
Fungi in the atmosphere are expected to be found mostly as fungal spores, although the relative
concentration of fungal spores and fungal hyphae fragments in air is unknown. Our results
showed that the number of sporulation-related functions was higher in air than the other
ecosystems (with the exception of snow and phyllosphere). While fungal hyphae are not
expected to be particularly resistant to extreme conditions such as UV radiation, fungal spores
are specifically produced to resist and survive overall adverse atmospheric conditions (Huang
and Hull, 2017). Their thick membrane and dehydrated nature make them particularly resistant
to abiotic atmospheric conditions such as UV radiation, oxidative stress, desiccation as well as
osmotic stress. **Fig 6** presents a conceptual model that could explain the higher ratio between
fungi and bacteria observed in air. During aerosolization and aerial transport, bacteria and fungi
might be under stress and might undergo a physical selection with the survival of the most
resistant cells to the adverse atmospheric conditions (*i.e.* UV radiation, desiccation *etc.*) and the
death of non-resistant cells. As fungi (and especially fungal spores) might be naturally more
resistant and adapted to atmospheric conditions than bacteria, we expect a larger decline of
bacterial cells compared to fungal cells and spores in air. This might have as a consequence an
increase in the ratio between fungi and bacteria compared to their non-atmospheric origins (*i.e.*
the surrounding ecosystems) (**Fig 6**).

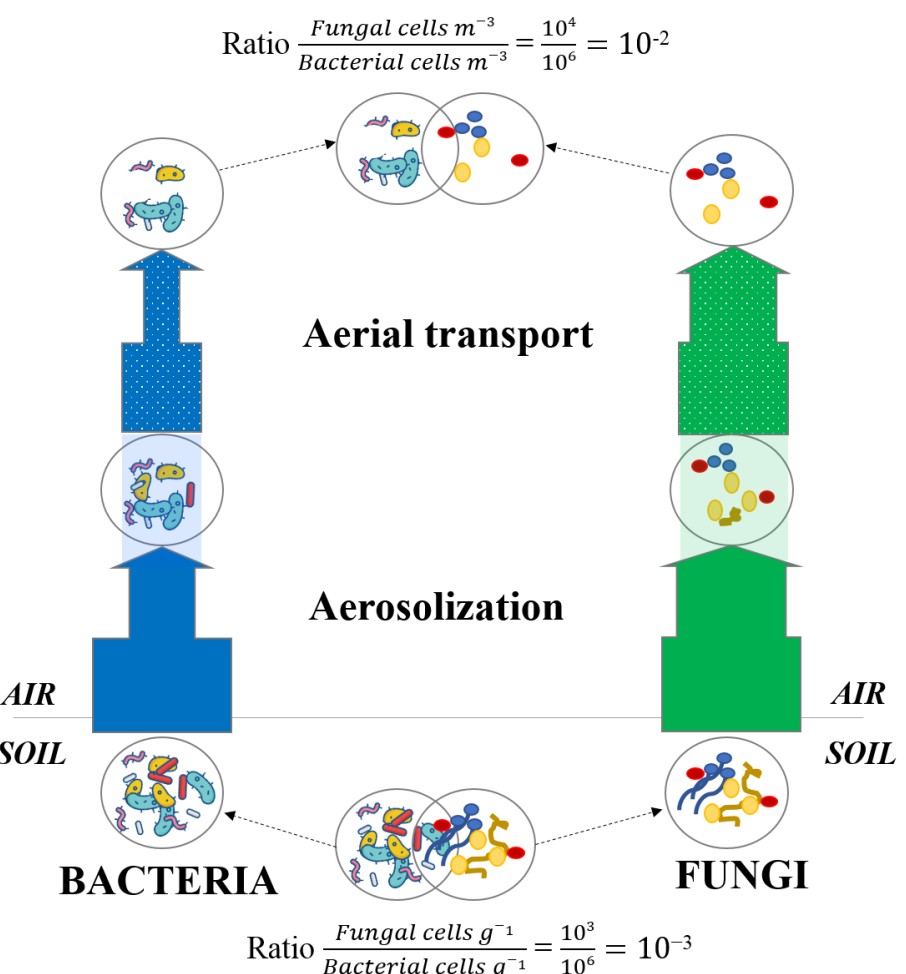


**Fig 6. Microbial cell loss due to atmospheric physical stress.** Conceptual model on the
microbial cell loss occurring during the aerosolization and aerial transport steps due to physical
selection. The thickness of the arrows represents the impact of the physical selection on both
bacterial and fungal cell loss (the more microbial cells survive the physical selection, the thicker
becomes the arrow). Approximate ratios are indicative and result from 16S rRNA and 18S
rRNA qPCR data on Côte Saint André soil samples (crop soil, France) and puy de Dôme air
samples (France; puy de Dôme landscape is mainly composed of croplands as shown in **Fig
S1**).
The high variability between the air sites and between air samples of the same site could be
explained by the variability in the inputs from the different surrounding landscapes. Our
previous paper showed that local inputs were the main sources of planetary boundary layer
microorganisms and that local meteorology (especially the wind direction) had a major impact
on the temporal variability of airborne microbial communities by affecting which of the





different local sources were upwind (Tignat-Perrier et al., 2019). Our results did not show a specific (metabolic) functional potential signature for the atmosphere, which was rather mainly driven by the surrounding landscapes. Our results are consistent with both a pre-metabolic adaptation of airborne microorganisms to the chemicals of the sources (*i.e.* surrounding landscapes) and a potential metabolic adaptation to these chemicals in the atmosphere.

Atmospheric chemistry is dependent on the underlying ecosystem chemistry since the main sources of atmospheric chemicals are Earth surface emissions. Yet, the oxidizing conditions of the atmosphere might lead to rapid transformations of atmospheric chemicals by photochemical reactions. These specific atmospheric chemical reactions (*i.e.* photochemical) produce species which, with the gases like $CH_4$, characterize the atmosphere ($O_3$, $H_2O_2$, OH *etc*.). Although some microbial strains from cloud water origin have been shown to metabolize and grow on culture medium in the presence of $H_2O_2$ (Vaïtilingom et al., 2013), radical species and their precursors are reactive compounds and might not easily serve as energy and carbon sources for microorganisms (Imlay, 2013). Our results on specific metabolic related functions showed that functions related to methane monooxygenase activity ($CH_4$ degradation) and hydrogen peroxide catabolism ($H_2O_2$ degradation) were present in air but not in higher proportion than in other ecosystems (**Fig S3**). Reactive compounds can cause oxidative stress to airborne microorganisms. In association to adverse physical conditions like UV radiation and desiccation, oxidative compounds might create more of a physical stress than provide a new metabolic source for airborne microorganisms. Laboratory investigations of cultivable microorganisms of an airborne origin showed the presence of particularly resistant strains under stressful conditions similar to the atmospheric ones (*i.e.* similar UV radiation levels; different oxidative conditions) (Joly et al., 2015; Yang et al., 2008). However, no study has shown whether these apparently adapted cells represented the majority of airborne microorganisms. Since the overall SEED functional classes included mainly metabolic functions, specific stress related functions using GO (Gene Ontology) terms were also evaluated. We observed that on average, air showed more stress-related functions (UV response, desiccation and oxidative stress response related functions) than the other ecosystems due to the higher concentration of fungi (relatively to bacteria) in air. Thus, when the annotated sequences were separated between sequences belonging to fungal and bacterial genomes, the bacterial and fungal sequences from air samples did not show a significantly higher concentration of stress-related functions compared to the samples coming from other ecosystems (**Fig 4, 5, Fig S4**).

Fungal genomes are expected to carry genes associated to global stress-related functions (*i.e.* UV radiation, desiccation, oxidative stress), because of the innate resistance of fungi especially fungal spores. These genes associated to global stress-related functions are likely acquired during sporulation formation and certainly do not result from adaptation of fungi in air. When studying genes coding more specific proteins that are not associated to spore resistance, such as lipoate synthase and chromosome plasmid partitioning protein ParA, that might play a role in oxidative stress (Allary et al., 2007; Bunik, 2003) and are more generally found in stress resistance and adaptability of microorganisms (Shoeb et al., 2012; Zhang et al., 2018), they were occasionally found in relatively high concentration in air samples (**Fig S3**). The detection of metagenomic sequences annotated as genes coding specific proteins in air samples remains difficult because of the low microbial biomass recovered. That is why we examined the presence and concentration of global functions (*i.e.* UV protection related functions, oxidative stress response related functions *etc*.) rather than specific functional genes.

The constant and large input of microbial cells to the planetary boundary layer and their relatively short residence time (a few hours to a few days based on a model assuming that microbial cells behave like non biological aerosols (Jaenicke, 1980)) might have hindered the observation of the potential adaptation (physical selection and/or microbial adaptation) of airborne microorganisms to the stressful atmospheric conditions and to the atmospheric




chemicals as discussed above. This issue might be addressed by investigating microbial functional potential in the free troposphere (preferentially high enough above the ground so as not to be influenced by the surface) where the microbial fluxes are smaller than in the planetary boundary layer and where microbial airborne residence time might last much longer than in the planetary boundary layer. This troposphere approach might help in determining the role of stress in the atmosphere and validate our conceptual model on the physical stress of microbial cells taking place during aerosolization and aerial transport selecting the resistant cells (**Fig 6**). Another explanation might be due to the metagenomic approach that allows to sample both living and dead cells. Aerosolization has been shown to be particularly stressful and even lethal for microorganisms (Alsved et al., 2018; Thomas et al., 2011). The functional potential from the dead cells in air might have a greater weight on the overall functional potential observed and lead to the dilution of the functional potential of the actual living cells that have adapted to atmospheric conditions. This might apply for both the overall functional potential discussed previously and the stress-related functions.

**Conclusion**

We conducted the first global comparative metagenomic analysis to characterize the microbial functional potential signature in the planetary boundary layer. Air samples showed no specific signature of microbial functional potential which was mainly correlated to the surrounding landscapes. However, air samples were characterized by a relatively high percentage of fungal sequences compared to the source ecosystems (soil, surface seawater *etc*.). The relatively higher concentrations of fungi in air drove the higher proportions of stress-related functions observed in air metagenomes. Fungal cells and specifically fungal spores are innately resistant entities well adapted to atmospheric conditions and which might survive better aerosolization and aerial transport than bacterial cells. Stress-related functions were present in airborne bacteria but rarely in higher concentrations compared to the bacterial communities in other ecosystems. However, the constant flux of microbial cells to the planetary boundary layer might have complicated the determination of a physical selection and/or microbial adaptation of airborne microorganisms, especially bacterial communities. Meta-omics investigations on air with a deeper sequencing are needed to confirm our results and explore the functionality of atmospheric microorganisms further.

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

**Competing interests.** The authors declare that they have no conflict of interest.

**Financial support**. This work was supported by the Agence Nationale de la Recherche
[ANR-15-CE01-0002–03 INHALE]; Région Auvergne-Rhône Alpes [ARC3 2016];
CAMPUS France [program XU GUANGQI] and the French Polar Institute IPEV [program
1028 and 399].
**Author contributions**. AD, CL and TMV designed the experiment. RTP, AD, AT and OM
conducted the sampling field campaign. RTP did the molecular biology, bioinformatics and
statistical analyses. RTP, AD, CL and TMV analyzed the results. RTP, TM, AD and CL wrote
the manuscript. All authors reviewed the manuscript.

**Acknowledgements**. The chemical analyses were performed at the IGE AirOSol platform. This
work was hosted by the following stations: Chacaltaya, Namco, puy de Dôme, Cape-Point, Pic-
du-Midi, Amsterdam-Island, Storm-Peak, Villum RS and we thank I.Jouvie, G.Hallar,
I.McCubbin, Benny and Jesper, B.Jensen, A.Nicosia, M.Ribeiro, L.Besaury, L.Bouvier,
M.Joly, I.Moreno, M.Rocca, F.Velarde for sampling and station management. We thank our
project partners: K.Sellegri, P.Amato, M.Andrade, Q.Zhang, C.Labuschagne and L.Martin, J.
Sonke. We thank R.Edwards, J. Schauer and C.Worley for lending their HV sampler. We thank
L.Pouilloux for computing assistance and maintenance of the Newton server.
**Data availability**. Sequences reported in this paper have been deposited in ftp://ftp-adn.ec-
lyon.fr/Tignat-Perrier_2020_air_metagen_INHALE/. A file has been attached explaining the
correspondence between file names and samples.