# Peer review of "Microbial functional signature in the atmospheric boundary layer"

_Biogeosciences, 2020_

## Referee Comment (RC1) · Anonymous Referee #1 · 23 Sep 2020

General comments: In this paper, the authors present results of metagenomic sequencing of air filter samples collected at nine different locations around the world with functional profiles for fungi and bacteria. The authors compare their data with data sets from various other ecosystems. The main aim of the study was to characterize the functional potential of the airborne community and to identify potential atmosphere-specific signatures. The results indicate that the functional potential of fungi and bacteria is not specific for the atmosphere but similar to the underlying ecosystem. The manuscript is well written but can be further improved as suggested below.

Specific comments: Real-Time PCR analysis: The authors give a method description for qPCR (starting L133) on air filter samples but the results for the air samples appear to be already published somewhere else as stated in L242/243 and caption of Table S4.

[Figure]

Table S4 gives also some qPCR results for soil samples. Where are these data from? Can the authors provide details in the method section about collecting/extracting/qPCR for the soil samples or a reference? To which air samples/location do these soil samples belong? Were they taken at the same time/location as air sampling was done? It is also not clear to which unit (L, m3, gram?) the gene copy numbers given in Table S4 refer and how the authors calculated the cell concentrations as stated in L133, L146, L243/244 and where these values can be found. The authors refer in L243 to Table S4 for cell concentration ratios, but this table only includes gene copy numbers and their ratios. Overall, the motivation for the qPCR seems not clear in terms of the purpose of the study. The qPCR is not mentioned in the abstract or introduction/motivation and the discussion is confusing as the term "gene copy numbers" seems to be used also as "cell concentrations". Given the multicopy nature of ribosomal genes and different copy numbers in different organisms this should be corrected in the text.

Metagenomic data analysis: The bioinformatic analysis appears to be focused only on fungi and bacteria, but there are also other microbial organisms such as e.g., Protozoa, Archaea, Algae in the atmosphere. Can the authors add information about the numbers of non-fungal and non-bacterial reads and explain how and why they were excluded although they are/many of them are microorganisms (see title!). Overall, it might be straighter to separate fungi and bacteria in all figures as they belong to different domains of life. For example, Fig 4a and Fig 5a seem not to provide any additional value to panles b and c, if panels a include only fungi plus bacteria but no other mircoorganisms.

2.2.3/2.3.2: How can the data be normalized to 10000 sequences (L225) when the filtering cut-off was 6000 sequences (L173) before?

Fig2: The numbers in the figure are hard to read. The grey for the air samples appears not to be in the figure. The order seems to be by % of fungal and bacterial sequences. To me it seems more useful to display the different sample groups (air, soil, water,..) so that one might be able to see trends. Does this figure display all sequences for

each of the samples/sample sets i.e., different total numbers of reads/sequences per sample/sample/set or is this figure based on rarefied sequences (6000, 2000)?

L204ff: The authors selected specific stress-related functions with the purpose to identify a specific atmospheric functional potential signature. Stresses like e.g., UV, desiccation, however, are not limited to the atmosphere. Also soil bacteria and microorganisms living on e.g., plant or building surfaces are exposed to these stresses. This might help to explain that the authors did not find a specific signature with their selected genes in the airborne fraction.

L322/323: The authors state that the methane mono-oxygenase-related functional proteins per 10000 sequences were only detectable when considering all sequences. As all sequences are the sum of fungal and bacterial sequences I wonder why they can only detect it when they sum up the sequences. If they have sequences in the sum, they must have had them for fungi and/or bacteria before. Can the authors clarify?

L470/471: Please correct the statement that concentration of fungal spores and fungal hyphae fragments in air are unknown. For example, numbers of spore and hyphae concentration can be found in Després et al., 2012 and references therein.

Després, V.R., Huffman, J.A., Burrows, S.M., Hoose, C., Safatov, A.S., Buryak, G., Fröhlich- Nowoisky, J., Elbert, W., Andreae, M.O., Pöschl, U., Jaenicke, R., 2012. Primary biological aerosol particles in the atmosphere: a review. Tellus B 64. http://dx.doi.org/10. 3402/tellusb.v64i0.15598

Table S1: site should be capitalized, abbreviations in first column should be explained; what is meant with "same hour"? There is no time information in this table.

Figure S1: This is a nice figure, but is only mentioned once and it seems not be used for discussion in the text. I suggest to consider this figure when discussing Fig.3, as it supports the results shown in Fig 3.

Figure S2: This figure is very hard to read as a lot of text overlaps

[Figure]

---

## Referee Comment (RC2) · Anonymous Referee #2 · 29 Sep 2020

General comments. This is an interesting study with a metagenomic analysis of a large amount of air samples from different parts of the world. The resulting metabolic signatures are attempted to correlate with other environments, both aqueous and terrestrial. The metabolic functions related to stress and resistance are analyzed and a predominance of those of fungi is observed. As a main conclusion, it is said that there is no specific atmospheric signature, but correlations with underlying ecosystems can be established. The great merit of this work is to develop a metagenomic study from air samples. In aerobiology, low biomass and low efficiency of the samplers make the application of molecular methodologies very complex. On the other hand, having been able to analyze samples in different parts of the world and different environments, is also a great success of this study, although that makes the results difficult to follow, es-

pecially when rpuntos about the underlying ecosystems, whose results do not appear clearly. Specific comments. - My main comment is regarding the underlying ecosystems. I have not found data on how the samples were sampled or how they were processed. It is also not clear to me where they came from. Some data are given in table S2, but it is not clear where they come from and why the samples are selected. On the other hand, are they exactly underlying ecosystems? That is, did the soils, sea waters or snow samples correspond to the sites where air samples were being collected? It is interesting to introduce soil samples far from the air sampling sites, but also to analyze those closer. The same applies to snow or sea. - On the other hand, much importance is given to fungi, but little is said about bacterial metabolic activities. - Didn't they consider the possibility of analyzing other eukaryotes or archaea as well? - L108-112. In aerobiology controls are vital. In this study, it is commented that control filters were taken based on previous studies (Dommergue et al. 2019). Can you give more details about these controls? Were they processed together with the rest of the samples? What results did they give? - L242. The qPCR data are said to be from a previous paper and are summarized in table S4, but it hardly presents any data, most of the sampling points do not appear. - In general, the legends and characters on the charts are difficult to read, especially those on the supplementary material.

---

## Author Comment (AC1) · 29 Sep 2020

We thank the Referee 1 for reviewing our paper and for the constructive comments that, we think, contributed to make the paper much more comprehensible especially the methodology.

-Comment 1: "The authors give a method description for qPCR (starting L133) on air filter samples but the results for the air samples appear to be already published somewhere else as stated in L242/243 and caption of Table S4. Table S4 gives also some qPCR results for soil samples. Where are these data from? Can the authors provide details in the method section about collecting/extracting/qPCR for the soil samples or a reference? To which air samples/location do these soil samples belong? Were they

taken at the same time/location as air sampling was done? It is also not clear to which unit (L, m3, gram?) the gene copy numbers given in Table S4 refer and how the authors calculated the cell concentrations as stated in L133, L146, L243/244 and where these values can be found. The authors refer in L243 to Table S4 for cell concentration ratios, but this table only includes gene copy numbers and their ratios. Overall, the motivation for the qPCR seems not clear in terms of the purpose of the study. The qPCR is not mentioned in the abstract or introduction/motivation and the discussion is confusing as the term "gene copy numbers" seems to be used also as "cell concentrations". Given the multicopy nature of ribosomal genes and different copy numbers in different organisms this should be corrected in the text."

Answer: Some qPCR results are actually already presented in our previous paper (Tignat-Perrier et al., 2019). Instead of giving all the qPCR methodology again, we modified the text and referenced the previous paper for method details (line 138). We made it clear in the Material and Methods section that the gene copy numbers were used as an approximation of the cell concentrations and added a reference, Louca et al., 2018, that explains why attempts to correct for metagenomic datasets are unproductive (while we agree of the multiple nature of ribosomal genes and different numbers in different organisms) (line 141). We agree that we forgot to give information on the soil samples and we gave details in the Material and Methods section and result section accordingly (line 129, line 138, line 225). The qPCR results on the soil samples, even partial, were used in comparison to the results obtained from the metagenomic data to see if they show similar results and/or the same trend. The soil is the Côte Saint André agricultural soil (France) that we also used for the metagenomic analyses. We agree that it would have been better to have samples from the surrounding landscape at each of our air sites, and we think that a study would be interesting to do to confirm our results and support the new conceptual model we proposed (Fig 6). We modified the text (line 129, line 138, line 225, line 453).

-Comment 2: "The bioinformatic analysis appears to be focused only on fungi and

bacteria, but there are also other microbial organisms such as e.g., Protozoa, Archaea, Algae in the atmosphere. Can the authors add information about the numbers of non-fungal and non-bacterial reads and explain how and why they were excluded although they are/many of them are microorganisms (see title!). Overall, it might be straighter to separate fungi and bacteria in all figures as they belong to different domains of life. For example, Fig 4a and Fig 5a seem not to provide any additional value to panles b and c, if panels a include only fungi plus bacteria but no other mircoorganisms. Âż

Answer: We agree that non-fungal eukaryotic organisms are certainly present in air and we think that looking at them would be very interesting. Still, our sampling method is not suitable to collect these bigger microorganisms. We used an impaction technique that mostly collected particulate matter whose diameter is inferior to 10 $\mu$m. We checked our data for non-fungal eukaryotic reads and they actually seem really low. As an example, a typical puy de Dôme sample (site relatively vegetated) showed 11900 over 12400 annotated sequences belonging to the Fungi reign. Non-fungal eukaryotic sequences and archaeal sequences were very low. We think that the panel showing bacterial and fungal sequences altogether ((a) panel) is still informative. The ratio between fungal and bacterial sequences is specifically high in air and this ratio drives the stress functions based results as shown when comparing the (a) panel including all sequences and the (b) and (c) panels including the fungal and bacterial sequences separately. Considering together bacterial and fungal reads is common place in metagenomic studies while it might be important to separate them, as evidenced here.

-Comment 3: "How can the data be normalized to 10000 sequences (L225) when the filtering cut-off was 6000 sequences (L173) before?"

Answer: We agree that it might have been not clear and we added information in the Material and Methods section in this regard (line 201). It was a deliberate choice to give the relative ratio of a specific function per 10000 sequences. It could have been per 100 sequences and we would have said that for example "0.2 % or 0.2 sequence

over 100 given sequences is related to this specific function". Still 0.2 sequence means less than 1 sequence and thus sounds odd. Thus we chose per 10000 sequences to have sequence numbers above 1 (0.2 sequences per 100 sequences would be 200 sequences per 10000 sequences).

-Comment 4: "Fig2: The numbers in the figure are hard to read. The grey for the air samples appears not to be in the figure. The order seems to be by % of fungal and bacterial sequences. To me it seems more useful to display the different sample groups (air, soil, water,..) so that one might be able to see trends. Does this figure display all sequences for each of the samples/sample sets i.e., different total numbers of reads/sequences per sample/sample/set or is this figure based on rarefied sequences (6000, 2000)?"

Answer: We increased the height of the numbers so that they would be more readable. We do not think that displaying only the average percentage of bacterial and fungal reads per ecosystem is helpful as the difference between sites (especially within soil and air sites) could be very large (due to the fact that the sites are more or less vegetated, affecting the percentage of fungal reads). We think that displaying the average combined with the very large standard deviation would not have been meaningful. The percentages were calculated using all sequences and not rarefied sequences. We added information in order to made it clear in the Material and Methods section (line 201).

-Comment 5: "The authors selected specific stress-related functions with the purpose to identify a specific atmospheric functional potential signature. Stresses like e.g., UV, desiccation, however, are not limited to the atmosphere. Also soil bacteria and microorganisms living on e.g., plant or building surfaces are exposed to these stresses. This might help to explain that the authors did not find a specific signature with their selected genes in the airborne fraction."

Answer: We agree that some stress-related functions we chose like UV and desiccation

could also be stresses experienced by surface microorganisms (i.e. microorganisms found on plant leaves, on soil surface, on sea surface etc.) and this is what we tried to explain in the discussion section line 494 and line 526.

-Comment 6: "The authors state that the methane mono-oxygenase-related functional proteins per 10000 sequences were only detectable when considering all sequences. As all sequences are the sum of fungal and bacterial sequences I wonder why they can only detect it when they sum up the sequences. If they have sequences in the sum, they must have had them for fungi and/or bacteria before. Can the authors clarify? Âż

Answer: Firstly, we used all reads and functionally annotated them (as explained line 172 in the Material and Methods section). In this case, the majority of the reads was related to bacteria and fungi (because of our sampling method), although some reads might belong to other reigns (for example Archaea, Protista. . .) as you have highlighted it in a previous comment. Secondly, we tried to separate bacterial reads from fungal reads (using Kraken, FindFungi and specific complete genome-based databases – line 183), then functionally annotated them separately. Thus, when taking in consideration all sequences, it is not exactly the sum of the bacterial and fungal sequences. We hope that the text in the Material and Methods section is clear enough.

-Comment 7: "Please correct the statement that concentration of fungal spores and fungal hyphae fragments in air are unknown. For example, numbers of spore and hyphae concentration can be found in Després et al., 2012 and references therein. Després, V.R., Huffman, J.A., Burrows, S.M., Hoose, C., Safatov, A.S., Buryak, G., Fröhlich- Nowoisky, J., Elbert, W., Andreae, M.O., Pöschl, U., Jaenicke, R., 2012. Primary biological aerosol particles in the atmosphere: a review. Tellus B 64. http://dx.doi.org/10. 3402/tellusb.v64i0.15598 Âż

Answer: We corrected the statement and made it clear that some numbers of spores and hyphae were measured and could be found in Després et al., 2012. We specified that the number of hyphae and spores, and thus the ratio between the two, has never

been measured at the same site (line 457).

-Comment 8: "Table S1: site should be capitalized, abbreviations in first column should be explained; what is meant with "same hour"? There is no time information in this table."

Answer: We changed the text accordingly. By "same hour", we meant that the collection was stopped exactly the same hour as it started (i.e. the sampling lasted exactly 7 days – even if the hour was not given because we thought it was not useful), but we removed this as it was not clear.

-Comment 9: "Figure S1: This is a nice figure, but is only mentioned once and it seems not be used for discussion in the text. I suggest to consider this figure when discussing Fig.3, as it supports the results shown in Fig 3."

Answer: We added a reference to Fig S1 when discussing Fig 3 (line 417).

-Comment 10: "Figure S2: This figure is very hard to read as a lot of text overlaps"

Answer: We agree that text overlaps and that not all the text is readable. We added this Fig S2 in SI so that readers can have indications on where the sites are situated on the multivariate analysis (as the main Fig 3 shows only the colors and not the text), even if we know that all the text could not be readable. We added colors (based on the "ecosystem") to make the reading easier in the case that the site names (i.e. the text) is not what interests the reader.

---

## Author Comment (AC2) · 1 Oct 2020

We thank the Referee 2 for reviewing our paper and the constructive comments.

We agree that it would have been very interesting to have samples from the ecosystems that directly underlie the air sampling. Our study is a preliminary comparative metagenomic study that gave a first insight on the microbial functional genes present in air compared to the ones commonly found in other environments, and we think that the next large-scale metagenomic study should include air samples as well as samples from the direct underlying environments (especially if we consider that short-range transport of microorganisms is more likely than long-range transport in air; Tignat-Perrier et al., 2019). This sampling, that should also include "simple" sites that are

surrounded by one type of environment over a long distance (such as a desert site or polar site), would provide interesting information on the aerosolization process (i.e. microbial populations that would be more likely aerosolized and thus found in air compared to others). Here, the non-air environmental samples were chosen on different public databases based on their ecosystem type and diversity within an ecosystem type (i.e. different forest ecosystems, different seas etc.) as well as based on the sequencing technique used (i.e. we wanted metagenomic datasets made from the Miseq, HiSeq and 454 technology). We downloaded the free-access datasets (database and reference number in the Table S2) and analyzed the fastq files in the same pipeline as our fastq files.

We agree that looking at non-fungal eukaryotic microorganisms in air would be very interesting. Still, our sampling method is not suitable to collect these bigger microorganisms. We used an impaction technique that mostly collected particulate matter whose diameter is inferior to 10 $\mu$m. We checked our data for non-fungal eukaryotic reads and they actually seem really low. As an example, a typical puy de Dôme sample (site relatively vegetated, in France) showed 11900 over 12400 annotated sequences belonging to the Fungi reign. Non-fungal eukaryotic sequences, as well as archaeal sequences were very low. Our study evidenced that fungi could be an important part of airborne microorganisms (quantitatively, compared to bacteria), and that is why our paper mostly focused on fungi.

Contamination in aerobiology is very hard to prevent but still, like the Referee 2 observed, very critical to control. Our sampling strategy, i.e. sampling a very large volume of air per sample using a high volume air sampler, made contamination effect, if existing, much less significant on our samples and allowed us to get a microbial biomass suitable for molecular analyses. The contamination level of our air sampling was accessed in Dommergue et al., 2019. The large air sampling done at different sites around the world allowed us to collect large samples on which we cut sub-samples to do chemical analyses (Dommergue et al., 2019 and Tignat-Perrier et al., 2019 and

2020) and analyses on the DNA such as qPCR and amplicon sequencing (16S rRNA gene and ITS; Tignat-Perrier et al., 2019 and 2020) as well as metagenomic sequencing (this paper). Controls (filters just put in the pump without functioning and transport filters that were transported in an aluminium fold and a plastic bag at the same time as the real samples but never opened) were collected and were processed at the same time as the real samples for the organic carbon concentration, qPCR and amplicon sequencing analyses. In the methodological paper (Dommergue et al., 2019), we gave results on the processing of the controls (organic carbon concentration and 16S rRNA gene qPCR) and identified the potential contamination in our air sampling. Only for the Antarctica site, the real samples showed DNA concentration so low (not detectable) that they could not be differentiated from the controls, and we decided to remove the Antarctica dataset altogether. For all sites, we verified that the DNA concentration on the controls was not detectable, and that the 16S rRNA gene qPCR and amplicon sequencing gave little and low-quality sequences compared to the corresponding samples, respectively. From Dommergue et al., 2019 we can read: "Except for the polar sites and CHC, the concentration of 16S rRNA gene copies in blank samples were < 0.3% that in the corresponding atmospheric samples. The blanks at CHC were up to 7% of the average number of copies in the atmospheric samples, due to the low concentrations of DNA sampled from air at this high altitude site. At both polar sites (DMC and Villum) the 16S rRNA gene concentrations were similar to controls, indicating very low biomass."

qPCR results based on our air samples are actually already presented in our previous paper (Tignat-Perrier et al., 2019) and in this paper we used only some qPCR results given as examples. Instead of giving all the qPCR methodology again, we modified the text and referenced the previous paper for method details (line 138). We agree that we forgot to give information on the soil samples (on which we did also qPCR analysis) and we gave details in the Material and Methods section and result section accordingly (line 129, line 138, line 225). The soil is the Côte Saint André agricultural soil (France) that we also used for the metagenomic analyses. Here, the partial qPCR

results were used in comparison to the results obtained from the metagenomic data to see if they show similar results and/or the same trend (especially regarding the ratio between fungi and bacteria). We agree that it would have been better to have samples (soil, sea samples etc.) from the surrounding landscape at each of our air sites (and do qPCR analysis but also metagenomic sequencing), and we think that a future study would be interesting to do to confirm our results and support the new conceptual model we proposed (Fig 6). We modified the text to give more information on the partial qPCR analysis and its purpose (line 129, line 138, line 225, line 453).

We agree that a few SI figures are difficult to read, especially the Fig S2. In the Fig S2 some text overlaps and not all the text is readable. We added this Fig S2 in SI so that readers can have indications on where the sites are situated on the multivariate analysis (as the main Fig 3 shows only the colors and not the text), even if we know that all the text could not be readable. We added colors (based on the ecosystem type) to make the reading easier in the case that the site names (i.e. the text) is not what interests the reader. For the SI tables, we agree that they are large tables with lot of text, but we tried to do everything to make them readable.

---

## Referee Comment (RC3) · Anonymous Referee #1 · 2 Oct 2020

I thank the authors for their replies and would like to comment again on the qPCR issue. The authors now explain the use of gene copy numbers as cell concentrations with unproductive corrections of metagenomic data sets. For metagenomic data, this appears true, but these data are from qPCR. A more solid estimation of cell concentration could be obtained when estimates of ribosomal copy numbers for fungi and bacteria are taken into account. By using the gene copy numbers directly as cell concentrations, the authors count one gene copy as one cell. But due to the multi copy nature of ribosomal genes, these gene copy number derived cell concentrations represent false high cell concentrations. Thus, it would be better and less miss-leading to the readers to use the words "gene copy numbers" instead of "cell concentrations" in the text, if the authors cannot find estimates of gene copy numbers for ribosomal genes of fungi and

bacteria in the literature to calculate more solid values for cell concentrations.

---

## Short Comment (SC1) · 2 Oct 2020

We thank the Referee 1 for the feedbacks and we appreciate the thoughtful discussion.

We changed the text, especially we used the words "gene copy numbers" instead of "cell concentrations". We agree that using cell concentration for qPCR gene copy number is misleading as one cell (bacterial or fungal) does not harbor one ribosomal gene copy. We could have chosen to divide the qPCR results by four if we stated that it was the average copy number of 16S rRNA gene per bacterial genome (that we think remains erroneous), yet our conclusions would not have changed as we used gene copy numbers for relative comparisons (we did not consider absolute values). We think that taking the multi copy nature of ribosomal genes (in qPCR analysis) into account

when considering overall microbial communities composed of thousands of bacterial and fungal species remains an important issue and we welcome all suggestions.

---

## Author Comment (AC3) · 5 Oct 2020

We thank the Referee 1 for the feedbacks and we appreciate the thoughtful discussion.

We changed the text, especially we used the words "gene copy numbers" instead of "cell concentrations". We agree that using cell concentration for qPCR gene copy num-ber is misleading as one cell (bacterial or fungal) does not harbor one riboso-mal genecopy. We could have chosen to divide the qPCR results by four if we stated that it wasthe average copy number of 16S rRNA gene per bacterial genome (that we think re-mains erroneous), yet our conclusions would not have changed as we used gene copynumbers for relative comparisons (we did not consider absolute val-ues). We thinkthat taking the multi copy nature of ribosomal genes (in qPCR analysis)

into accountC1 BGDInteractivecommentPrinter-friendly versionDiscussion paperwhen considering overall microbial communities composed of thousands of bacterialand fungal species remains an important issue and we welcome all suggestions.

———————————————————

---

## Author Response (AR1)

The authors would like to thank the Associate Editor for handling and reviewing our manuscript, as well as the Referees for reviewing our paper and their feedbacks.

We took into account all Referees and Associate Editor comments and modified the text and SI file accordingly. The changes are highlighted in grey.

Below our additional responses regarding Associate Editor specific comments:

-We increased the size of the numbers in the Figure 2 in the way that they do not appear blurry while being not too large (as they are just here to detail the y axis). We could increase again the size if necessary.

-We completed the Figure 3 legend.

-We added information on the air samplers used in the Material and Methods section, and added a column in the Table S1 to indicate which sampler was used for each sample. We have photos of the samplers that we could add in the SI file.

-We removed the word "preliminary" in the Abstract. We wanted to say that it is the first metagenomic study comparing air metagenomes with metagenomes coming from other ecosystems, but we agree that our sentence might have been confusing.

-We modified the text regarding the qPCR data and purpose in the different sections. We hope that we made it clearer.

[revised manuscript text omitted]

**all, bacterial or fungal sequences.**

**Fig S3**. **Proportion of sequences annotated as hydrogen peroxide catabolic process related**

**functional proteins as well as proteins potentially implicated in stress resistance in the**

**metagenomes**.

**Fig S4**. **Proportion of sequences annotated as UV protection and desiccation response related**

**functional proteins in the metagenomes**.

**3 Supplementary tables**

**Table S1**. **Air sample collection characteristics.**

**Table S2**. **Characteristics of the metagenomes.**

**Table S3**. **Functional richness and evenness averaged per site.**

**Table S4**. **Ratio between fungal and bacterial cell concentration in air and soil.**

**Table S5**. **Dominant SEED functions in the metagenomes.**

**1 Supplementary Materials**

[Figure]

**Fig S1**. **Surrounding landscapes of the air sampling sites.** The proportions of different landscapes within a perimeter of 50 km have been calculated based on the land cover MODIS approach. AMS: Amsterdam-Island, CAP: Cape-Point, STN: Station Nord, CHC: Chacaltaya, NAM: Namco, STP: Storm Peak, PDM: Pic-du-Midi, PDD: puy de Dôme, GRE: Grenoble.

**2 Supplementary Figures**

[Figure]

**Fig S2**. **Distribution of the samples based on the microbial functional profile when considering all, bacterial or fungal sequences.** PCoA analysis of the Bray-Curtis dissimilarity matrix based on the functional potential structure of each site. All sequences (**a**), bacterial sequences (**b**) and fungal sequences (**c**) have been used for functional annotation. For the site including several metagenomes, the average profile was calculated. Colors indicate the ecosystems in which the sites belong to.

[Figure]

**Fig S3**. **Proportion of sequences annotated as hydrogen peroxide catabolic process related functional proteins as well as proteins potentially implicated in stress resistance in the metagenomes**. Average number of sequences annotated as proteins implicated in the hydrogen peroxide catabolic process (**a**, left) per 10000 annotated sequences from (**i**) all sequences, (**ii**) fungal sequences and (**iii**) bacterial sequences per site, and average number of hits of lipoate synthase (**b**) and chromosome plasmid partitioning protein ParA (**c**) per 10000 annotated sequences from all sequences per site. Colors indicate the ecosystems in which the sites belong to. For the sites including several metagenomes, the standard deviation was added.

[Figure]

**Fig S4**. **Proportion of sequences annotated as UV protection and desiccation response related functional proteins in the metagenomes**. Average number of sequences annotated as proteins implicated in the UV protection (**a**, left) and desiccation response (**b**, right) per 10000 annotated sequences from (**i**) all sequences, (**ii**) fungal sequences and (**iii**) bacterial sequences per site. Colors indicate the ecosystems in which the sites belong to. For the sites including several metagenomes, the standard deviation was added.

**3 Supplementary tables**

**Table S1**. **Air sample collection characteristics.** Standardized collected air volume and sampling starting date of each air sample that we collected for this study. AMS: Amsterdam Island, CAP: Cape Point, CHC: Chacaltaya, GRE: Grenoble, NAM: Namco, PDD: Puy de Dôme, PDM: Pic du Midi, STN: Station Nord, STP: Storm Peak

| Site | Sample name | Standardized collected air volume ($m^3$) | High volume air sampler | Sampling starting date (ending date: exactly 7 days after) (month/day/year) |
|------|-------------|-------------------------------------------|-------------------------|------------------------------------------------------------------------------|
| AMS | AMS_10/09/2016 | 5232 | Digitel PM10 head + pump | 10/09/16 |
| AMS | AMS_08/10/2016 | 4449 | Digitel PM10 head + pump | 08/10/16 |
| AMS | AMS_21/10/2016 | 5059 | Digitel PM10 head + pump | 10/21/16 |
| CAP | CAP_21/10/2016 | 4679 | Digitel DA77 sampler | 10/21/16 |
| CAP | CAP_28/10/2016 | 545 | Digitel DA77 sampler | 10/28/16 |
| CHC | CHC_23/09/2016 | 1148 | Digitel PM10 head + pump | 09/23/16 |
| CHC | CHC_21/10/2016 | 1154 | Digitel PM10 head + pump | 10/21/16 |
| CHC | CHC_28/10/2016 | 1220 | Digitel PM10 head + pump | 10/28/16 |
| CHC | CHC_01/07/2016 | 1284 | Digitel PM10 head + pump | 01/07/16 |
| CHC | CHC_08/07/2016 | 1284 | Digitel PM10 head + pump | 08/07/16 |
| CHC | CHC_15/07/2016 | 1289 | Digitel PM10 head + pump | 07/15/16 |
| CHC | CHC_12/08/2016 | 1160 | Digitel PM10 head + pump | 12/08/16 |
| CHC | CHC_19/08/2016 | 1158 | Digitel PM10 head + pump | 08/19/16 |
| CHC | CHC_02/09/2016 | 1156 | Digitel PM10 head + pump | 02/09/16 |
| GRE | GRE_03/07/2017 | 4688 | Digitel PM10 head + pump | 03/07/17 |
| GRE | GRE_10/07/2017 | 4717 | Digitel PM10 head + pump | 10/07/17 |
| GRE | GRE_17/07/2017 | 4677 | Digitel PM10 head + pump | 07/17/17 |
| GRE | GRE_24/07/2017 | 4718 | Digitel PM10 head + pump | 07/24/17 |
| GRE | GRE_31/07/2017 | 4665 | Digitel PM10 head + pump | 07/31/17 |
| GRE | GRE_07/08/2017 | 4762 | Digitel PM10 head + pump | 07/08/17 |
| GRE | GRE_14/08/2017 | 4729 | Digitel PM10 head + pump | 08/14/17 |
| GRE | GRE_21/08/2017 | 4707 | Digitel PM10 head + pump | 08/21/17 |
| GRE | GRE_04/09/2017 | 4742 | Digitel PM10 head + pump | 04/09/17 |
| NAM | NAM_17/05/2017 | 5511 | 2131 Laowin chinese sampler | 05/17/17 |
| NAM | NAM_25/05/2017 | 5503 | 2131 Laowin chinese sampler | 05/25/17 |
| NAM | NAM_02/06/2017 | 5513 | 2131 Laowin chinese sampler | 02/06/17 |
| NAM | NAM_13/06/2017 | 4218 | 2131 Laowin chinese sampler | 06/13/17 |
| NAM | NAM_20/06/2017 | 5418 | 2131 Laowin chinese sampler | 06/20/17 |

| | | | | |
|---|---|---|---|---|
| NAM | NAM_29/06/2017 | 5415 | 2131 Laowin chinese sampler | 06/29/17 |
| NAM | NAM_07/07/2017 | 5483 | 2131 Laowin chinese sampler | 07/07/17 |
| NAM | NAM_14/07/2017 | 5413 | 2131 Laowin chinese sampler | 07/14/17 |
| NAM | NAM_21/07/2017 | 5465 | 2131 Laowin chinese sampler | 07/21/17 |
| PDD | PDD_07/06/2017 | 8761 | Digitel PM10 head + pump | 06/07/17 |
| PDD | PDD_14/06/2017 | 8360 | Digitel PM10 head + pump | 06/14/17 |
| PDD | PDD_21/06/2017 | 8672 | Digitel PM10 head + pump | 06/21/17 |
| PDD | PDD_28/06/2017 | 9012 | Digitel PM10 head + pump | 06/28/17 |
| PDD | PDD_02/08/2017 | 7399 | Digitel PM10 head + pump | 08/02/17 |
| PDD | PDD_09/08/2017 | 9926 | Digitel PM10 head + pump | 08/09/17 |
| PDD | PDD_30/05/2017 | 10232 | Digitel PM10 head + pump | 05/30/17 |
| PDD | PDD_12/07/2017 | 8713 | Digitel PM10 head + pump | 07/12/17 |
| PDD | PDD_19/07/2017 | 8620 | Digitel PM10 head + pump | 07/19/17 |
| PDD | PDD_26/07/2017 | 8664 | Digitel PM10 head + pump | 07/26/17 |
| PDM | PDM_20/06/2016 | 9664 | TISCH TE-5170V sampler | 06/20/16 |
| PDM | PDM_29/06/2016 | 6803 | TISCH TE-5170V sampler | 06/29/16 |
| PDM | PDM_12/07/2016 | 7550 | TISCH TE-5170V sampler | 12/07/16 |
| PDM | PDM_19/07/2016 | 8040 | TISCH TE-5170V sampler | 07/19/16 |
| PDM | PDM_26/07/2016 | 7794 | TISCH TE-5170V sampler | 07/26/16 |
| PDM | PDM_02/08/2016 | 8103 | TISCH TE-5170V sampler | 02/08/16 |
| PDM | PDM_09/08/2016 | 7747 | TISCH TE-5170V sampler | 08/09/16 |
| PDM | PDM_16/08/2016 | 8100 | TISCH TE-5170V sampler | 08/16/16 |
| PDM | PDM_23/08/2016 | 7956 | TISCH TE-5170V sampler | 08/23/16 |
| PDM | PDM_13/09/2016 | 7931 | TISCH TE-5170V sampler | 09/13/16 |
| PDM | PDM_20/09/2016 | 7853 | TISCH TE-5170V sampler | 09/20/16 |
| PDM | PDM_06/09/2016 | 7867 | TISCH TE-5170V sampler | 06/09/16 |
| PDM | PDM_16/08/2016 | 8100 | TISCH TE-5170V sampler | 08/16/16 |
| STN | STN_27/03/2017 | 5153 | Digitel DA80 sampler | 03/27/17 |
| STN | STN_15/05/2017 | 5246 | Digitel DA80 sampler | 05/15/17 |
| STP | STP_14/07/2017 | 11213 | TISCH PM10 sampler | 07/14/17 |
| STP | STP_21/07/2017 | 9333 | TISCH PM10 sampler | 07/21/17 |
| STP | STP_28/07/2017 | 5702 | TISCH PM10 sampler | 07/28/17 |
| STP | STP_11/08/2017 | 5702 | TISCH PM10 sampler | 08/11/17 |
| STP | STP_18/08/2017 | 5702 | TISCH PM10 sampler | 08/18/17 |
| STP | STP_25/08/2017 | 5702 | TISCH PM10 sampler | 08/25/17 |

Table S2. **Characteristics of the metagenomes.** Number of samples, ecosystem, sequencing technology, database and accession number, number of sequences per sample (mean + standard deviation), percentage of fungal and bacterial sequences per site and percentage of annotated sequences (mean + standard deviation) per site.

| Site | Country/ Ocean | Information on the site | Number of samples | Ecosystem | Sequencing technology | Database and reference numbers or study | Total sequence number | Annotated sequence number by eggNOG-Mapper | Fungi-associated sequences | Percentage of fungi-associated sequences over total read number | Percentage of fungi-associated sequences over fungi-and bacteria-associated sequence number | Percentage of fungi-associated sequences annotated by eggNOG-Mapper | Bacteria-associated sequence number | Percentage of bacteria-associated sequences over total sequence number | Percentage of bacteria-associated sequences over fungi-and bacteria-associated sequence number | Percentage of bacteria-associated sequences annotated by eggNOG- Mapper |
|---|---|---|---|---|---|---|---|---|---|---|---|---|---|---|---|---|
| Air Amsterdam-Island (AMS) | Sub-Antarctica | marine, remote | 3 | air | MiSeq | present study | 97881 +- 93551 | 17676 +- 15935 | 4152 +- 4089 | 4.1 +- 0.2 | 71 +- 4 | 60 +- 2 | 1670 +- 1549 | 1.7 +- 0.4 | 29 +- 4 | 57 +- 2 |
| Air Beijing | China | urban | 2 | air | HiSeq | mgm4516366.3, mgm4516459.3 | 2248590 +- 177298 | 141849 +- 9576 | 250843 +- 9161 | 11 +- 3.2 | 52 +- 10 | 17 +- 1 | 226290 +- 8208 | 10.1 +- 1.2 | 48 +- 10 | 39 +- 1 |
| Air Cape-Point (CAP) | South-Africa | coastal | 2 | air | MiSeq | present study | 90043 +- 6341 | 20479 +- 1447 | 7227 +- 5972 | 7.8 +- 6.1 | 56 +- 29 | 50 +- 12 | 4530 +- 1286 | 5.1 +- 1.8 | 44 +- 29 | 61 +- 1 |
| Air Chacaltaya (CHC) | Bolivia | high-altitude mountain peak | 9 | air | MiSeq | present study | 103239 +- 54187 | 32699 +- 18131 | 3479 +- 2580 | 3.5 +- 1.2 | 27 +- 24 | 61 +- 7 | 11113 +- 6411 | 10.2 +- 2 | 73 +- 24 | 67 +- 2 |
| Air Grenoble (GRE) | France | urban | 9 | air | MiSeq | present study | 248064 +- 158109 | 42853 +- 30690 | 24234 +- 15561 | 9.7 +- 0.5 | 79 +- 10 | 48 +- 3 | 7082 +- 8061 | 2.7 +- 1.6 | 21 +- 10 | 59 +- 5 |
| Air New York indoors (indoors_NY) | USA | indoors | 4 | air | 454 | SRR1000232, SRR1000254, SRR999213, SRR999215 | 400997 +- 49680 | 126245 +- 12742 | 36858 +- 16604 | 9.2 +- 4.1 | 52 +- 17 | 47 +- 8 | 32035 +- 10923 | 8.3 +- 3.7 | 48 +- 17 | 70 +- 9 |
| Air Namco (NAM) | China | high-altitude plateau, semi-arid | 9 | air | MiSeq | present study | 149952 +- 92976 | 48012 +- 36340 | 2958 +- 1910 | 2.1 +- 1.1 | 19 +- 12 | 68 +- 9 | 15901 +- 13188 | 10 +- 2.6 | 81 +- 12 | 69 +- 2 |
| Air New York (NY) | USA | urban, coastal | 6 | air | 454 | SRR1000260, SRR1000269, SRR999217, SRR999218, SRR999219, SRR999220 | 521791 +- 277049 | 99566 +- 51023 | 85350 +- 41529 | 20 +- 11.6 | 56 +- 8 | 46 +- 41 | 69161 +- 38301 | 18.1 +- 14.5 | 44 +- 8 | 37 +- 47 |
| Air Puy-de-Dôme (PDD) | France | continental, moutain peak | 10 | air | MiSeq | present study | 396666 +- 364681 | 65304 +- 68592 | 25029 +- 32432 | 5.8 +- 2.5 | 62 +- 16 | 50 +- 6 | 13112 +- 12079 | 3.9 +- 3 | 38 +- 16 | 56 +- 4 |

| Sample | Country | Habitat | n | Type | Platform | Source | | | | | | | | | | |
|---|---|---|---|---|---|---|---|---|---|---|---|---|---|---|---|---|
| Air Pic-du-Midi (PDM) | France | high-altitude moutain peak | 13 | air | MiSeq | present study | 186766 +- 197396 | 33016 +- 31653 | 8676 +- 8233 | 5 +- 2.1 | 50 +- 10 | 54 +- 4 | 8115 +- 7687 | 5.5 +- 2.6 | 50 +- 10 | 63 +- 5 |
| Air San-Diego | USA | urban coastal | 2 | air | 454 | SRR999211, SRR999212 | 781206 +- 65608 | 229544 +- 3651 | 5960 +- 4678 | 0.8 +- 0.7 | 36 +- 1 | 44 +- 3 | 10318 +- 7888 | 1.4 +- 1.1 | 64 +- 1 | 54 +- 13 |
| Air Station-Nord (STN) | Greenland | polar | 2 | air | MiSeq | present study | 23463 +- 24385 | 5935 +- 5528 | 1276 +- 1702 | 3.6 +- 3.5 | 24 +- 18 | 59 +- 38 | 2460 +- 2606 | 10.2 +- 0.5 | 76 +- 18 | 65 +- 3 |
| Air Storm-Peak (STP) | USA | high-altitude mountain peak | 6 | air | MiSeq | present study | 469168 +- 242715 | 111530 +- 58582 | 99110 +- 56113 | 20.7 +- 2.6 | 88 +- 4 | 53 +- 2 | 12559 +- 7185 | 2.8 +- 1.1 | 12 +- 4 | 59 +- 5 |
| Human feces Sydney | Australia | human feces | 1 | feces | HiSeq | mgm4675774.3 | 2111825 +- NA | 503328 +- NA | 28357 +- NA | 1.3 +- NA | 3 +- NA | 29 +- NA | 929682 +- NA | 44 +- NA | 97 +- NA | 35 +- NA |
| Hydrothermal vent Pacific | Pacific Ocean | deep sea water | 1 | hydrothermal vent | 454 | mgm4481541.3 | 758485 +- NA | 409294 +- NA | 12501 +- NA | 1.6 +- NA | 9 +- NA | 56 +- NA | 128294 +- NA | 16.9 +- NA | 91 +- NA | 84 +- NA |
| Leaf surface Israel | Israel | leaf surface | 1 | phyllosphere | HiSeq | mgm4534773.3 | 12272440 +- NA | 3274840 +- NA | 460612 +- NA | 3.8 +- NA | 37 +- NA | 23 +- NA | 784441 +- NA | 6.4 +- NA | 63 +- NA | 70 +- NA |
| Seawater Atlantic sea | North Atlantic Ocean | surface sea water | 1 | seawater | MiSeq | mgm4719942.3 | 1198007 +- NA | 301519 +- NA | 54904 +- NA | 4.6 +- NA | 36 +- NA | 11 +- NA | 99510 +- NA | 8.3 +- NA | 64 +- NA | 51 +- NA |
| Seawater Balearic sea | Balearic Sea | surface sea water | 1 | seawater | MiSeq | mgm4719938.3 | 878884 +- NA | 531765 +- NA | 50995 +- NA | 5.8 +- NA | 36 +- NA | 60 +- NA | 89238 +- NA | 10.2 +- NA | 64 +- NA | 88 +- NA |
| Seawater Celtic sea | Celtic Sea | surface sea water | 1 | seawater | MiSeq | mgm4719941.3 | 1702779 +- NA | 283691 +- NA | 510855 +- NA | 11.4 +- NA | 53 +- NA | 18 +- NA | 112212 +- NA | 6.6 +- NA | 47 +- NA | 55 +- NA |
| Deep abyss USA | USA | deep sea water | 1 | seawater | 454 | mgm4668304.3 | 2016153 +- NA | 330380 +- NA | 1895 +- NA | 0.1 +- NA | 2 +- NA | 68 +- NA | 90930 +- NA | 4.5 +- NA | 98 +- NA | 75 +- NA |
| Seawater Greenland sea | Greenland Sea | surface sea water | 1 | seawater | MiSeq | mgm4719947.3 | 1358477 +- NA | 687315 +- NA | 79494 +- NA | 5.9 +- NA | 28 +- NA | 51 +- NA | 200245 +- NA | 14.7 +- NA | 72 +- NA | 77 +- NA |
| Seawater Indian sea | Indian Sea | surface sea water | 1 | seawater | MiSeq | mgm4719994.3 | 126564 +- NA | 56086 +- NA | 2527 +- NA | 2 +- NA | 11 +- NA | 55 +- NA | 21431 +- NA | 16.9 +- NA | 89 +- NA | 89 +- NA |
| Seawater Irish sea | Irish Sea | surface sea water | 1 | seawater | MiSeq | mgm4719940.3 | 1362228 +- NA | 538286 +- NA | 27051 +- NA | 2 +- NA | 28 +- NA | 21 +- NA | 69952 +- NA | 5.1 +- NA | 72 +- NA | 70 +- NA |
| Seawater Mediterranean sea | Mediterranean Sea, Easterm bassin | surface sea water | 1 | seawater | MiSeq | mgm4719936.3 | 1241549 +- NA | 686395 +- NA | 17384 +- NA | 1.4 +- NA | 11 +- NA | 43 +- NA | 133889 +- NA | 10.8 +- NA | 89 +- NA | 87 +- NA |
| River water Colorado | USA | river water | 1 | riverwater | HiSeq | mgm4628878.3 | 1392059 +- NA | 575616 +- NA | 39553 +- NA | 2.8 +- NA | 19 +- NA | 31 +- NA | 169972 +- NA | 12.2 +- NA | 81 +- NA | 76 +- NA |
| River sediments West Virginia | USA | river sediments | 1 | sediments | HiSeq | mgm4589537.3 | 2072338 +- NA | 569631 +- NA | 58672 +- NA | 2.8 +- NA | 26 +- NA | 28 +- NA | 166079 +- NA | 8 +- NA | 74 +- NA | 71 +- NA |
| Sediments Peru | Peru | seafloor sediments | 2 | sediments | 454 | mgm4440960.3, mgm4459940.3 | 126239 +- 35030 | 3606 +- 829 | 1780 +- 2323 | 1.2 +- 1.5 | 53 +- 47 | 13 +- 16 | 539 +- 18 | 0.4 +- 0.1 | 47 +- 47 | 33 +- 11 |
| Sediments Pond | France | shallow pond sediments | 4 | sediments | MiSeq | our lab; Sanchez-Cid *et al.*, under review | 44236 +- 17409 | 15568 +- 7046 | 1450 +- 991 | 3.2 +- 1.2 | 9 +- 3 | 80 +- 5 | 16704 +- 12966 | 36.7 +- 18.1 | 91 +- 3 | 56 +- 8 |
| Snow Svalbard | Norway | fresh snow, artic | 7 | snow | MiSeq | our lab; Bergk-Pinto *et al.*, under review | 226368 +- 70313 | 49428 +- 23023 | 3788 +- 1542 | 1.7 +- 0.6 | 18 +- 6 | 61 +- 12 | 18469 +- 8699 | 8.5 +- 3.2 | 82 +- 6 | 54 +- 5 |
| Agricultural soil France | France | agricultural soil | 3 | soil | MiSeq | mgm4705012.3, mgm4697958.3, mgm4697957.3 | 8209393 +- 2836681 | 5913635 +- 1326693 | 194398 +- 108165 | 2.3 +- 0.5 | 61 +- 3 | 60 +- 13 | 129460 +- 84783 | 1.5 +- 0.5 | 39 +- 3 | 65 +- 11 |

| Name | Country | Soil type | n | Env | Platform | Reference | | | | | | | | | | |
|---|---|---|---|---|---|---|---|---|---|---|---|---|---|---|---|---|
| Soil Cote Saint Andre | France | agricultural soil | 6 | soil | MiSeq | our lab; Sanchez-Cid et al., under review | 174898 +- 80968 | 54841 +- 22225 | 1638 +- 640 | 1 +- 0.1 | 9 +- 1 | 82 +- 2 | 16806 +- 6846 | 9.8 +- 0.6 | 91 +- 1 | 67 +- 1 |
| Soil CraibStone | Scotland | agricultural soil | 5 | soil | MiSeq | our lab; Sanchez-Cid et al., under review | 128815 +- 82837 | 41175 +- 22413 | 1452 +- 953 | 1.1 +- 0.1 | 9 +- 1 | 78 +- 4 | 15472 +- 11906 | 11.5 +- 1.2 | 91 +- 1 | 65 +- 1 |
| Cropland UK | United Kingdom | cropland | 2 | soil | MiSeq | mgm4781436.3, mgm4781437.3 | 485163 +- 163475 | 304642 +- 104503 | 9970 +- 3861 | 2 +- 0.1 | 13 +- 0 | 88 +- NA | 66040 +- 25119 | 13.5 +- 0.6 | 87 +- 0 | 95 +- 0 |
| Forest soil Baltimore | USA | temperate deciduous broadleaf forest soil | 1 | soil | MiSeq | mgm4819073.3 | 4600481 +- NA | 959764 +- NA | 34207 +- NA | 0.7 +- NA | 12 +- NA | 62 +- NA | 260826 +- NA | 5.7 +- NA | 88 +- NA | 79 +- NA |
| Grassland USA | South-Africa | tropical grassland | 1 | soil | MiSeq | mgm4819072.3 | 2519738 +- NA | 638149 +- NA | 22974 +- NA | 0.9 +- NA | 8 +- NA | 63 +- NA | 279551 +- NA | 11.1 +- NA | 92 +- NA | 79 +- NA |
| Grassland USA | USA | temperate grassland | 2 | soil | MiSeq | mgm4623641.3, mgm4623640.3 | 12195227 +- 1436683 | 697967 +- 122932 | 1753640 +- 1903 | 14.5 +- 1.7 | 42 +- 1 | 2 +- 0 | 2381163 +- 99911 | 19.6 +- 1.5 | 58 +- 1 | 13 +- 2 |
| Soil Lucknow India | India | soil | 1 | soil | 454 | mgm4461840.3 | 1187505 +- NA | 658023 +- NA | 53911 +- NA | 4.5 +- NA | 14 +- NA | 91 +- NA | 322160 +- NA | 27.1 +- NA | 86 +- NA | 91 +- NA |
| Soil Nevada | USA | soil | 1 | soil | 454 | mgm4451106.3 | 1248623 +- NA | 725892 +- NA | 29880 +- NA | 2.4 +- NA | 8 +- NA | 84 +- NA | 326929 +- NA | 26.2 +- NA | 92 +- NA | 91 +- NA |
| Plant soil USA | USA | soil | 1 | soil | HiSeq | mgm4767414.3 | 17632266 +- NA | 1425603 +- NA | 253827 +- NA | 1.4 +- NA | 15 +- NA | 33 +- NA | 1473019 +- NA | 8.4 +- NA | 85 +- NA | 52 +- NA |
| Prairie Kansas | USA | prairie soil | 1 | soil | MiSeq | mgm4477804.3 | 5348832 +- NA | 343702 +- NA | 27412 +- NA | 0.5 +- NA | 9 +- NA | 51 +- NA | 270464 +- NA | 5.1 +- NA | 91 +- NA | 56 +- NA |
| Soil PuertoRico | Puerto Rico | subtropical forest | 1 | soil | 454 | mgm4446153.3 | 725275 +- NA | 452063 +- NA | 10926 +- NA | 1.5 +- NA | 11 +- NA | 88 +- NA | 85868 +- NA | 11.8 +- NA | 89 +- NA | 95 +- NA |
| Rhizosphere Amazonia | Brazil | tropical broadleaf forest | 1 | soil | HiSeq | mgm4723911.3 | 8884491 +- NA | 1415017 +- NA | 2075 +- NA | 0 +- NA | 0 +- NA | 52 +- NA | 1027815 +- NA | 11.6 +- NA | 100 +- NA | 60 +- NA |
| Soil South-Africa | South-Africa | tropical grassland | 1 | soil | MiSeq | mgm4819068.3 | 2757834 +- NA | 759329 +- NA | 25123 +- NA | 0.9 +- NA | 6 +- NA | 64 +- NA | 414056 +- NA | 15 +- NA | 94 +- NA | 79 +- NA |
| Shrubland California | USA | shrubland | 1 | soil | MiSeq | mgm4806895.3 | 2213724 +- NA | 47591 +- NA | 3742 +- NA | 0.2 +- NA | 1 +- NA | 11 +- NA | 243528 +- NA | 11 +- NA | 98 +- NA | 7 +- NA |
| Shrubland Sudan | Sudan | shrubland | 1 | soil | MiSeq | mgm4806896.3 | 185966 +- NA | 1169 +- NA | 1181 +- NA | 0.6 +- NA | 18 +- NA | 7 +- NA | 5381 +- NA | 2.9 +- NA | 82 +- NA | 11 +- NA |

**Table S3**. **Functional richness and evenness averaged per site.** Functional richness and evenness after rarefaction per site, based on the SEED functional classes. For site including several samples, the mean and standard deviation have been calculated.

| Site | Ecosystem | All sequences | | | | Fungal sequences | | | | Bacterial sequences | | | |
|---|---|---|---|---|---|---|---|---|---|---|---|---|---|
| | | Number of annotated sequences using Diamond and MEGAN6 | Rarefaction | Functional richness after rarefaction | Functional evenness after rarefaction | Number of annotated sequences using Diamond and MEGAN6 | Rarefaction | Functional richness after rarefaction | Functional evenness after rarefaction | Number of annotated sequences using Diamond and MEGAN6 | Rarefaction | Functional richness after rarefaction | Functional evenness after rarefaction |
| air Amsterdam-Island (AMS) | air | 3927 +- 3321 | 1737 +- 456 | 1087 +- 554 | 0.94 +- 0.02 | 81 +- 80 | 81 +- 80 | 66 +- 58 | 0.99 +- 0.01 | 554 +- 480 | 360 +- 193 | 270 +- 197 | 0.96 +- 0.01 |
| air Beijing | air | 180196 +- 11408 | 2000 +- 0 | 4060 +- 112 | 0.86 +- 0 | 5960 +- 214 | 500 +- 0 | 1129 +- 92 | 0.89 +- 0 | 82004 +- 5643 | 500 +- 0 | 2835 +- 58 | 0.87 +- 0 |
| air Cape-Point (CAP) | air | 8176 +- 4856 | 2000 +- 0 | 1634 +- 337 | 0.93 +- 0.02 | 211 +- 15 | 211 +- 15 | 162 +- 6 | 0.97 +- 0 | 1890 +- 726 | 500 +- 0 | 739 +- 192 | 0.95 +- 0.01 |
| air Chacaltaya (CHC) | air | 15853 +- 8907 | 1848 +- 456 | 2062 +- 714 | 0.92 +- 0.02 | 380 +- 219 | 346 +- 175 | 223 +- 109 | 0.96 +- 0.02 | 5268 +- 3052 | 467 +- 99 | 1142 +- 461 | 0.93 +- 0.02 |
| air Grenoble (GRE) | air | 5765 +- 6870 | 1802 +- 297 | 1256 +- 700 | 0.94 +- 0.02 | 412 +- 382 | 308 +- 156 | 235 +- 153 | 0.97 +- 0.01 | 2193 +- 2949 | 445 +- 86 | 658 +- 528 | 0.96 +- 0.02 |
| air indoors New York (indoors_NY) | air | 32135 +- 11235 | 2000 +- 0 | 3302 +- 299 | 0.91 +- 0.01 | 1546 +- 802 | 500 +- 0 | 697 +- 206 | 0.95 +- 0.01 | 10067 +- 4782 | 500 +- 0 | 2183 +- 387 | 0.93 +- 0.01 |
| air Namco (NAM) | air | 23081 +- 19276 | 2000 +- 0 | 2280 +- 478 | 0.91 +- 0.01 | 596 +- 495 | 381 +- 114 | 287 +- 136 | 0.95 +- 0.02 | 7600 +- 6515 | 500 +- 0 | 1300 +- 372 | 0.92 +- 0.02 |
| air New York (NY) | air | 5481 +- 4324 | 1639 +- 561 | 1384 +- 849 | 0.89 +- 0.04 | 286 +- 231 | 275 +- 217 | 150 +- 109 | 0.91 +- 0.07 | 769 +- 622 | 362 +- 205 | 446 +- 335 | 0.94 +- 0.02 |
| air Puy-de-Dôme (PDD) | air | 11053 +- 9757 | 1976 +- 75 | 1700 +- 775 | 0.93 +- 0.02 | 656 +- 748 | 300 +- 198 | 297 +- 239 | 0.96 +- 0.03 | 4277 +- 4138 | 500 +- 0 | 989 +- 617 | 0.94 +- 0.03 |
| air Pic-du-Midi (PDM) | air | 9422 +- 8988 | 1769 +- 490 | 1575 +- 778 | 0.94 +- 0.02 | 363 +- 354 | 267 +- 185 | 218 +- 163 | 0.98 +- 0.02 | 3252 +- 3366 | 460 +- 101 | 832 +- 511 | 0.95 +- 0.01 |
| air San Diego | air | 14573 +- 8176 | 2000 +- 0 | 2021 +- 81 | 0.9 +- 0 | 184 +- 191 | 184 +- 191 | 91 +- 66 | 0.96 +- 0.04 | 1737 +- 1841 | 468 +- 46 | 628 +- 429 | 0.95 +- 0.01 |
| air Station-Nord (STN) | air | 2863 +- 2408 | 1580 +- 594 | 956 +- 547 | 0.95 +- 0.02 | 111 +- 111 | 111 +- 111 | 88 +- 81 | 0.98 +- 0.01 | 1089 +- 1085 | 411 +- 127 | 486 +- 400 | 0.96 +- 0.01 |
| air Storm-Peak (STP) | air | 11763 +- 7684 | 2000 +- 0 | 1865 +- 519 | 0.92 +- 0.01 | 973 +- 537 | 476 +- 58 | 392 +- 131 | 0.94 +- 0.02 | 3757 +- 3006 | 500 +- 0 | 971 +- 409 | 0.94 +- 0.02 |
| human feces Sydney | feces | 560641 +- NA | 2000 +- NA | 2802 +- NA | 0.82 +- NA | 3591 +- NA | 500 +- NA | 317 +- NA | 0.61 +- NA | 304338 +- NA | 500 +- NA | 2557 +- NA | 0.87 +- NA |
| leaf surface Israel | phyllosphere | 1042866 +- NA | 2000 +- NA | 4292 +- NA | 0.87 +- NA | 10644 +- NA | 500 +- NA | 1336 +- NA | 0.85 +- NA | 247373 +- NA | 500 +- NA | 3165 +- NA | 0.88 +- NA |
| river water USA | river water | 295902 +- NA | 2000 +- NA | 3550 +- NA | 0.87 +- NA | 6142 +- NA | 500 +- NA | 1001 +- NA | 0.89 +- NA | 76857 +- NA | 500 +- NA | 2497 +- NA | 0.87 +- NA |
| seawater Balearic sea | seawater | 340618 +- NA | 2000 +- NA | 2957 +- NA | 0.87 +- NA | 17554 +- NA | 500 +- NA | 1825 +- NA | 0.9 +- NA | 55185 +- NA | 500 +- NA | 1823 +- NA | 0.86 +- NA |
| seawater Celtic sea | seawater | 335790 +- NA | 2000 +- NA | 3325 +- NA | 0.87 +- NA | 11736 +- NA | 500 +- NA | 1831 +- NA | 0.91 +- NA | 41271 +- NA | 500 +- NA | 1964 +- NA | 0.87 +- NA |
| deep abyss USA | seawater | 333284 +- NA | 2000 +- NA | 3649 +- NA | 0.87 +- NA | 1006 +- NA | 500 +- NA | 390 +- NA | 0.92 +- NA | 43055 +- NA | 500 +- NA | 2590 +- NA | 0.9 +- NA |
| seawater Greenland sea | seawater | 417826 +- NA | 2000 +- NA | 3223 +- NA | 0.88 +- NA | 21407 +- NA | 500 +- NA | 2164 +- NA | 0.91 +- NA | 97826 +- NA | 500 +- NA | 2376 +- NA | 0.87 +- NA |
| hydrothermal Vent Pacific Ocean | hydrothermal vent | 217796 +- NA | 2000 +- NA | 3621 +- NA | 0.87 +- NA | 3855 +- NA | 500 +- NA | 950 +- NA | 0.9 +- NA | 62974 +- NA | 500 +- NA | 2533 +- NA | 0.89 +- NA |
| seawater Indian sea | seawater | 40507 +- NA | 2000 +- NA | 2178 +- NA | 0.9 +- NA | 799 +- NA | 500 +- NA | 476 +- NA | 0.96 +- NA | 12001 +- NA | 500 +- NA | 1183 +- NA | 0.93 +- NA |
| seawater Irish sea | seawater | 287629 +- NA | 2000 +- NA | 3283 +- NA | 0.88 +- NA | 2394 +- NA | 500 +- NA | 662 +- NA | 0.89 +- NA | 32848 +- NA | 500 +- NA | 1843 +- NA | 0.86 +- NA |

| | | | | | | | | | | | | | |
|---|---|---|---|---|---|---|---|---|---|---|---|---|---|
| seawater Mediterranean sea | seawater | 381180 +- NA | 2000 +- NA | 3375 +- NA | 0.87 +- NA | 3999 +- NA | 500 +- NA | 898 +- NA | 0.87 +- NA | 74727 +- NA | 500 +- NA | 2112 +- NA | 0.88 +- NA |
| seawater North Atlantic Ocean | seawater | 206085 +- NA | 2000 +- NA | 3143 +- NA | 0.87 +- NA | 2956 +- NA | 500 +- NA | 771 +- NA | 0.9 +- NA | 35702 +- NA | 500 +- NA | 1663 +- NA | 0.89 +- NA |
| sediments Peru | sediments | 6348 +- 1251 | 2000 +- 0 | 1138 +- 7 | 0.92 +- 0 | 13 +- 4 | 13 +- 4 | 13 +- 4 | 1 +- 0 | 133 +- 66 | 133 +- 66 | 90 +- 33 | 0.97 +- 0.02 |
| sediments Pond | sediments | 10569 +- 5084 | 2000 +- 0 | 1791 +- 252 | 0.92 +- 0.01 | 647 +- 422 | 441 +- 106 | 367 +- 149 | 0.97 +- 0.01 | 7038 +- 5091 | 500 +- 0 | 1364 +- 392 | 0.93 +- 0.01 |
| river sediments West Virginia | sediments | 315551 +- NA | 2000 +- NA | 3869 +- NA | 0.87 +- NA | 10660 +- NA | 500 +- NA | 1223 +- NA | 0.88 +- NA | 74774 +- NA | 500 +- NA | 2381 +- NA | 0.88 +- NA |
| snow Svalbard | snow | 26069 +- 15249 | 2000 +- 0 | 2243 +- 498 | 0.91 +- 0.01 | 648 +- 360 | 418 +- 152 | 329 +- 143 | 0.96 +- 0.02 | 8702 +- 4544 | 500 +- 0 | 1317 +- 356 | 0.92 +- 0.02 |
| agricultural soil France | soil | 907295 +- 258019 | 2000 +- 0 | 764 +- 47 | 0.72 +- 0.02 | 8707 +- 6045 | 442 +- 116 | 129 +- 74 | 0.68 +- 0.1 | 7044 +- 2095 | 500 +- 0 | 118 +- 55 | 0.49 +- 0.05 |
| soil Cote Saint Andre | soil | 34947 +- 16517 | 2000 +- 0 | 2552 +- 263 | 0.9 +- 0.01 | 783 +- 355 | 491 +- 14 | 350 +- 92 | 0.94 +- 0.01 | 8680 +- 3716 | 500 +- 0 | 1418 +- 191 | 0.91 +- 0.01 |
| soil CraibStone | soil | 27629 +- 18784 | 2000 +- 0 | 2406 +- 327 | 0.91 +- 0.01 | 668 +- 473 | 465 +- 46 | 336 +- 127 | 0.95 +- 0.02 | 7980 +- 6183 | 500 +- 0 | 1346 +- 289 | 0.92 +- 0.01 |
| cropland UK | soil | 122625 +- 44684 | 2000 +- 0 | 3490 +- 132 | 0.87 +- 0 | 5106 +- 1863 | 500 +- 0 | 1001 +- 112 | 0.89 +- 0.02 | 34986 +- 13329 | 500 +- 0 | 2254 +- 103 | 0.89 +- 0.01 |
| forest soil Baltimore | soil | 606468 +- NA | 2000 +- NA | 3998 +- NA | 0.86 +- NA | 33130 +- 27346 | 500 +- 0 | 1438 +- 259 | 0.84 +- 0.02 | 334568 +- 288964 | 500 +- 0 | 2518 +- 4 | 0.86 +- 0 |
| soil Puerto Rico | soil | 170277 +- NA | 2000 +- NA | 3607 +- NA | 0.86 +- NA | 6380 +- NA | 500 +- NA | 971 +- NA | 0.88 +- NA | 45321 +- NA | 500 +- NA | 2229 +- NA | 0.87 +- NA |
| grassland SA | soil | 191711 +- 271120 | 1000 +- 1414 | 2093 +- 2959 | 0.43 +- 0.61 | 14252 +- 9262 | 500 +- 0 | 1216 +- 199 | 0.85 +- 0.01 | 305911 +- 260034 | 500 +- 0 | 3147 +- 93 | 0.9 +- 0.02 |
| grassland USA | soil | 926515 +- 144620 | 2000 +- 0 | 3496 +- 45 | 0.86 +- 0 | 23892 +- 4105 | 500 +- 0 | 1539 +- 35 | 0.84 +- 0 | 242586 +- 44860 | 500 +- 0 | 2332 +- 42 | 0.86 +- 0 |
| soil lucknow India | soil | 303263 +- NA | 2000 +- NA | 3971 +- NA | 0.88 +- NA | 19077 +- NA | 500 +- NA | 1357 +- NA | 0.87 +- NA | 145062 +- NA | 500 +- NA | 2707 +- NA | 0.89 +- NA |
| soil Nevada | soil | 305066 +- NA | 2000 +- NA | 3528 +- NA | 0.87 +- NA | 12931 +- NA | 500 +- NA | 1324 +- NA | 0.85 +- NA | 147232 +- NA | 500 +- NA | 2465 +- NA | 0.89 +- NA |
| plant soil USA | soil | 1839508 +- NA | 2000 +- NA | 3610 +- NA | 0.87 +- NA | 65330 +- NA | 500 +- NA | 1663 +- NA | 0.84 +- NA | 612710 +- NA | 500 +- NA | 2491 +- NA | 0.87 +- NA |
| prairie Kansas | soil | 475235 +- NA | 2000 +- NA | 3393 +- NA | 0.86 +- NA | 11087 +- NA | 500 +- NA | 1155 +- NA | 0.85 +- NA | 119659 +- NA | 500 +- NA | 2282 +- NA | 0.87 +- NA |
| rhizosphere Amazonia | soil | 1061420 +- NA | 2000 +- NA | 3330 +- NA | 0.81 +- NA | 337 +- NA | 337 +- NA | 157 +- NA | 0.91 +- NA | 412216 +- NA | 500 +- NA | 1912 +- NA | 0.79 +- NA |
| soil South-Africa | soil | 464748 +- NA | 2000 +- NA | 4239 +- NA | 0.88 +- NA | 8883 +- NA | 500 +- NA | 1092 +- NA | 0.85 +- NA | 187173 +- NA | 500 +- NA | 3252 +- NA | 0.91 +- NA |
| shrubland California | soil | 65837 +- NA | 2000 +- NA | 2693 +- NA | 0.87 +- NA | 298 +- NA | 298 +- NA | 126 +- NA | 0.92 +- NA | 14541 +- NA | 500 +- NA | 1757 +- NA | 0.89 +- NA |
| shrubland Sudan | soil | 1751 +- NA | 1751 +- NA | 864 +- NA | 0.96 +- NA | 77 +- NA | 77 +- NA | 68 +- NA | 0.99 +- NA | 727 +- NA | 500 +- NA | 437 +- NA | 0.97 +- NA |

**Table S4**. **Ratio between 16S and 18S rRNA gene copy numbers in air and soil.** qPCR on the 16s rRNA gene and on the 18S rRNA gene on air and soil samples, and ratio between these qPCRs. Means and standard deviations were calculated on three (Cote Saint André), nine (Amsterdam-Island and Namco) and ten (Grenoble) samples. qPCR results for the air samples have already been presented in Tignat-Perrier et al., 2019.

| | qPCR 18S rRNA gene number | qPCR 16S rRNA gene number | Ratio qPCR16S/qPCR18S |
|---|---|---|---|
| **AIR SAMPLES** | | | |
| NAM (Namco) | $4.97 \times 10^{3} \pm 3.44 \times 10^{3}$ | $3.56 \times 10^{6} \pm 3.01 \times 10^{6}$ | 716 |
| GRE (Grenoble) | $5.28 \times 10^{4} \pm 3.61 \times 10^{4}$ | $1.20 \times 10^{6} \pm 9.38 \times 10^{5}$ | 23 |
| AMS (Amsterdam-Island) | $7.51 \times 10^{3} \pm 6.96 \times 10^{3}$ | $1.49 \times 10^{5} \pm 9.17 \times 10^{4}$ | 20 |
| **SOIL SAMPLES** | | | |
| Côte Saint André | $1.13 \times 10^{3} \pm 2.9 \times 10^{2}$ | $3.70 \times 10^{6} \pm 1.9 \times 10^{6}$ | 3265 |

**Table S5**. **Dominant SEED functions in the metagenomes.** Top 50 of the SEED functions observed in the air samples (mean +/- standard deviation) considering all the sequences (*i.e.* bacterial and fungal sequences).

| Function | air_AMS | air_Beijing | air_CAP | air_CHC | air_GRE | air_IndoorsNY | air_NAM | air_NY | air_PDD | air_PDM | air_SanDiego | air_STN | air_STP |
|---|---|---|---|---|---|---|---|---|---|---|---|---|---|
| \5-FCL-like protein\"" | 2.01 +/- 0.66 | 2.04 +/- 0.12 | 1.6 +/- 0.13 | 1.9 +/- 0.13 | 1.84 +/- 0.33 | 1.62 +/- 0.08 | 1.97 +/- 0.15 | 1.53 +/- 0.54 | 1.68 +/- 0.32 | 1.78 +/- 0.37 | 1.41 +/- 0.12 | 1.47 +/- 0.24 | 1.81 +/- 0.31 |
| \Long-chain-fatty-acid--CoA ligase (EC 6.2.1.3)\"" | 2.3 +/- 0.09 | 1.35 +/- 0.3 | 1.29 +/- 0.47 | 1.46 +/- 0.25 | 1.62 +/- 0.27 | 1.17 +/- 0.2 | 1.51 +/- 0.12 | 1.12 +/- 0.53 | 1.79 +/- 1.2 | 1.58 +/- 0.22 | 0.91 +/- 0.02 | 2.01 +/- 0.45 | 1.86 +/- 0.42 |
| \TonB-dependent receptor\"" | 1.3 +/- 0.44 | 0.36 +/- 0.03 | 1.02 +/- 0.4 | 0.94 +/- 0.14 | 1.07 +/- 0.46 | 0.91 +/- 0.15 | 0.98 +/- 0.12 | 0.72 +/- 0.12 | 0.99 +/- 0.51 | 1.03 +/- 0.28 | 0.45 +/- 0.04 | 1.06 +/- 0.52 | 0.83 +/- 0.27 |
| \3-oxoacyl-[acyl-carrier protein] reductase (EC 1.1.1.100)\"" | 0.65 +/- 0.28 | 0.55 +/- 0.02 | 0.75 +/- 0.01 | 0.9 +/- 0.11 | 0.8 +/- 0.26 | 0.62 +/- 0.06 | 1.04 +/- 0.11 | 0.56 +/- 0.17 | 0.75 +/- 0.25 | 0.79 +/- 0.24 | 0.45 +/- 0.05 | 0.84 +/- 0.09 | 0.91 +/- 0.21 |
| \COG2363\"" | 0.53 +/- 0.3 | 0.5 +/- 0.02 | 0.58 +/- 0.14 | 0.51 +/- 0.08 | 0.58 +/- 0.16 | 0.56 +/- 0.07 | 0.49 +/- 0.11 | 0.26 +/- 0.2 | 0.54 +/- 0.13 | 0.55 +/- 0.13 | 0.56 +/- 0.26 | 0.48 +/- 0.07 | 0.45 +/- 0.05 |
| \Aldehyde dehydrogenase (EC 1.2.1.3)\"" | 0.43 +/- 0.21 | 0.4 +/- 0.03 | 0.44 +/- 0.1 | 0.31 +/- 0.04 | 0.58 +/- 0.13 | 0.29 +/- 0.07 | 0.39 +/- 0.06 | 0.68 +/- 1.11 | 0.42 +/- 0.2 | 0.35 +/- 0.17 | 0.29 +/- 0.04 | 0.25 +/- 0.23 | 0.47 +/- 0.09 |
| \Adenylate cyclase (EC 4.6.1.1)\"" | 0.26 +/- 0.18 | 0.17 +/- 0.02 | 0.18 +/- 0.11 | 0.36 +/- 0.15 | 0.27 +/- 0.18 | 0.29 +/- 0.06 | 0.79 +/- 0.07 | 0.25 +/- 0.24 | 0.41 +/- 0.23 | 0.46 +/- 0.14 | 0.23 +/- 0.06 | 0.53 +/- 0.02 | 0.32 +/- 0.1 |
| \Beta-galactosidase (EC 3.2.1.23)\"" | 0.15 +/- 0.06 | 0.23 +/- 0.05 | 0.34 +/- 0.03 | 0.34 +/- 0.15 | 0.25 +/- 0.14 | 0.22 +/- 0.05 | 0.3 +/- 0.06 | 1.41 +/- 2.13 | 0.28 +/- 0.16 | 0.3 +/- 0.21 | 0.22 +/- 0.08 | 0.18 +/- 0.25 | 0.35 +/- 0.07 |
| \DNA-directed RNA polymerase beta' subunit (EC 2.7.7.6)\"" | 0.58 +/- 0.13 | 0.53 +/- 0.03 | 0.39 +/- 0.07 | 0.38 +/- 0.07 | 0.3 +/- 0.13 | 0.15 +/- 0.04 | 0.38 +/- 0.04 | 0.09 +/- 0.08 | 0.51 +/- 0.42 | 0.39 +/- 0.11 | 0.32 +/- 0.11 | 0.29 +/- 0.17 | 0.39 +/- 0.07 |
| \Aspartate aminotransferase (EC 2.6.1.1)\"" | 0.53 +/- 0.08 | 0.39 +/- 0.02 | 0.35 +/- 0.08 | 0.38 +/- 0.07 | 0.35 +/- 0.13 | 0.43 +/- 0.07 | 0.33 +/- 0.05 | 0.4 +/- 0.32 | 0.35 +/- 0.14 | 0.34 +/- 0.09 | 0.29 +/- 0.01 | 0.37 +/- 0.28 | 0.33 +/- 0.03 |
| \Cobalt-zinc-cadmium resistance protein CzcA\"" | 0.17 +/- 0.1 | 0.3 +/- 0.08 | 0.2 +/- 0.01 | 0.36 +/- 0.06 | 0.26 +/- 0.13 | 0.49 +/- 0.14 | 0.26 +/- 0.04 | 0.72 +/- 0.24 | 0.4 +/- 0.18 | 0.38 +/- 0.15 | 0.24 +/- 0.01 | 0.63 +/- 0.41 | 0.22 +/- 0.04 |
| \DNA topoisomerase I (EC 5.99.1.2)\"" | 0.07 +/- 0.07 | 0.17 +/- 0.01 | 0.08 +/- 0.05 | 0.12 +/- 0.05 | 0.08 +/- 0.03 | 0.12 +/- 0.04 | 0.16 +/- 0.04 | 3.08 +/- 4.05 | 0.14 +/- 0.11 | 0.1 +/- 0.06 | 0.16 +/- 0.01 | 0.27 +/- 0.1 | 0.1 +/- 0.03 |
| \High-affinity carbon uptake protein Hat/HatR\"" | 0.29 +/- 0.04 | 0.16 +/- 0.09 | 0.09 +/- 0.05 | 0.19 +/- 0.09 | 1.27 +/- 0.94 | 0.17 +/- 0.06 | 0.35 +/- 0.18 | 0.31 +/- 0.18 | 0.23 +/- 0.15 | 0.14 +/- 0.09 | 0.08 +/- 0.04 | 0.12 +/- 0.17 | 0.32 +/- 0.19 |
| \Beta-lactamase\"" | 0.47 +/- 0.18 | 0.15 +/- 0.01 | 0.28 +/- 0.04 | 0.29 +/- 0.12 | 0.23 +/- 0.12 | 0.26 +/- 0.06 | 0.41 +/- 0.09 | 0.62 +/- 0.64 | 0.24 +/- 0.1 | 0.41 +/- 0.11 | 0.26 +/- 0 | 0.2 +/- 0.16 | 0.3 +/- 0.05 |
| \DNA-directed RNA polymerase beta subunit (EC 2.7.7.6)\"" | 0.39 +/- 0.19 | 0.49 +/- 0.03 | 0.35 +/- 0.1 | 0.35 +/- 0.07 | 0.24 +/- 0.11 | 0.15 +/- 0.06 | 0.34 +/- 0.09 | 0.04 +/- 0.04 | 0.44 +/- 0.48 | 0.3 +/- 0.09 | 0.31 +/- 0.12 | 0.51 +/- 0.26 | 0.3 +/- 0.04 |
| \Butyryl-CoA dehydrogenase (EC 1.3.99.2)\"" | 0.34 +/- 0.22 | 0.29 +/- 0.04 | 0.31 +/- 0.15 | 0.3 +/- 0.13 | 0.36 +/- 0.16 | 0.19 +/- 0.05 | 0.33 +/- 0.04 | 0.21 +/- 0.1 | 0.3 +/- 0.17 | 0.3 +/- 0.09 | 0.19 +/- 0.02 | 0.36 +/- 0.23 | 0.35 +/- 0.09 |
| \Aspartyl-tRNA(Asn) amidotransferase subunit A (EC 6.3.5.6)\"" | 0.16 +/- 0.15 | 0.08 +/- 0 | 0.04 +/- 0.06 | 0.1 +/- 0.14 | 0.04 +/- 0.05 | 0.04 +/- 0.03 | 0.06 +/- 0.03 | 3.02 +/- 4.05 | 0.04 +/- 0.03 | 0.06 +/- 0.05 | 0.11 +/- 0.02 | 0.11 +/- 0.09 | 0.07 +/- 0.03 |
| \FIG039061: hypothetical protein related to heme utilization\"" | 0.28 +/- 0.19 | 0.34 +/- 0.02 | 0.36 +/- 0.12 | 0.29 +/- 0.12 | 0.32 +/- 0.09 | 0.33 +/- 0.04 | 0.26 +/- 0.05 | 0.17 +/- 0.14 | 0.31 +/- 0.13 | 0.3 +/- 0.11 | 0.37 +/- 0.04 | 0.2 +/- 0.09 | 0.34 +/- 0.07 |
| \DNA polymerase III alpha subunit (EC 2.7.7.7)\"" | 0.32 +/- 0.11 | 0.39 +/- 0 | 0.47 +/- 0.15 | 0.34 +/- 0.06 | 0.27 +/- 0.13 | 0.24 +/- 0.02 | 0.36 +/- 0.04 | 0.11 +/- 0.09 | 0.27 +/- 0.09 | 0.32 +/- 0.2 | 0.29 +/- 0.06 | 0.17 +/- 0.12 | 0.24 +/- 0.09 |
| \D-3-phosphoglycerate dehydrogenase (EC 1.1.1.95)\"" | 0.48 +/- 0.09 | 0.26 +/- 0 | 0.27 +/- 0.06 | 0.29 +/- 0.08 | 0.3 +/- 0.13 | 0.29 +/- 0.02 | 0.3 +/- 0.06 | 0.1 +/- 0.08 | 0.29 +/- 0.12 | 0.3 +/- 0.12 | 0.27 +/- 0.14 | 0.49 +/- 0.41 | 0.25 +/- 0.08 |
| \3-ketoacyl-CoA thiolase (EC 2.3.1.16)\"" | 0.43 +/- 0.06 | 0.41 +/- 0.03 | 0.24 +/- 0.1 | 0.28 +/- 0.04 | 0.31 +/- 0.17 | 0.19 +/- 0.02 | 0.3 +/- 0.04 | 0.16 +/- 0.1 | 0.25 +/- 0.11 | 0.29 +/- 0.14 | 0.23 +/- 0.12 | 0.37 +/- 0.16 | 0.34 +/- 0.1 |
| \Arylsulfatase (EC 3.1.6.1)\"" | 0.61 +/- 0.09 | 0.11 +/- 0.03 | 1.17 +/- 0.93 | 0.17 +/- 0.03 | 0.16 +/- 0.09 | 0.08 +/- 0.02 | 0.28 +/- 0.09 | 0.09 +/- 0.05 | 0.15 +/- 0.11 | 0.19 +/- 0.11 | 2.98 +/- 0.08 | 0.2 +/- 0.09 | 0.12 +/- 0.05 |
| \Enoyl-CoA hydratase (EC 4.2.1.17)\"" | 0.31 +/- 0.1 | 0.24 +/- 0 | 0.24 +/- 0.05 | 0.31 +/- 0.12 | 0.27 +/- 0.11 | 0.26 +/- 0.07 | 0.34 +/- 0.07 | 0.16 +/- 0.06 | 0.24 +/- 0.06 | 0.3 +/- 0.11 | 0.21 +/- 0.04 | 0.46 +/- 0.04 | 0.27 +/- 0.04 |
| \Acetyl-coenzyme A synthetase (EC 6.2.1.1)\"" | 0.15 +/- 0.07 | 0.38 +/- 0.01 | 0.36 +/- 0.21 | 0.35 +/- 0.09 | 0.29 +/- 0.09 | 0.21 +/- 0.04 | 0.27 +/- 0.04 | 0.27 +/- 0.16 | 0.24 +/- 0.11 | 0.28 +/- 0.07 | 0.2 +/- 0.12 | 0.23 +/- 0.04 | 0.31 +/- 0.03 |
| \Copper-translocating P-type ATPase (EC 3.6.3.4)\"" | 0.12 +/- 0.05 | 0.35 +/- 0.05 | 0.18 +/- 0.04 | 0.2 +/- 0.09 | 0.24 +/- 0.17 | 0.62 +/- 0.21 | 0.15 +/- 0.04 | 0.7 +/- 0.23 | 0.28 +/- 0.18 | 0.23 +/- 0.14 | 0.19 +/- 0.14 | 0.29 +/- 0.17 | 0.17 +/- 0.08 |
| \diguanylate cyclase/phosphodiesterase (GGDEF & EAL domains) with PAS/PAC sensor(s)\"" | 0.11 +/- 0.14 | 0.14 +/- 0.01 | 0.12 +/- 0.07 | 0.25 +/- 0.08 | 0.24 +/- 0.14 | 0.26 +/- 0.02 | 0.38 +/- 0.06 | 0.13 +/- 0.08 | 0.3 +/- 0.18 | 0.33 +/- 0.14 | 0.16 +/- 0.08 | 0.5 +/- 0.1 | 0.28 +/- 0.08 |
| \Chaperone protein DnaK\"" | 0.29 +/- 0.2 | 0.34 +/- 0.05 | 0.26 +/- 0.06 | 0.24 +/- 0.06 | 0.3 +/- 0.17 | 0.17 +/- 0.03 | 0.28 +/- 0.03 | 0.16 +/- 0.1 | 0.3 +/- 0.14 | 0.29 +/- 0.12 | 0.29 +/- 0.09 | 0.37 +/- 0.21 | 0.23 +/- 0.03 |

| Protein | | | | | | | | | | | | | |
|---|---|---|---|---|---|---|---|---|---|---|---|---|---|
| UDP-glucose 4-epimerase (EC 5.1.3.2) | 0.27 +/- 0.12 | 0.22 +/- 0.01 | 0.22 +/- 0.08 | 0.35 +/- 0.09 | 0.24 +/- 0.06 | 0.19 +/- 0.04 | 0.4 +/- 0.06 | 0.09 +/- 0.08 | 0.22 +/- 0.1 | 0.29 +/- 0.09 | 0.17 +/- 0.05 | 0.41 +/- 0.15 | 0.23 +/- 0.12 |
| Excinuclease ABC subunit A | 0.11 +/- 0.11 | 0.35 +/- 0.02 | 0.4 +/- 0.14 | 0.33 +/- 0.12 | 0.21 +/- 0.07 | 0.22 +/- 0.04 | 0.28 +/- 0.07 | 0.2 +/- 0.12 | 0.18 +/- 0.11 | 0.37 +/- 0.27 | 0.34 +/- 0.03 | 0.2 +/- 0.03 | 0.2 +/- 0.05 |
| Aconitate hydratase (EC 4.2.1.3) | 0.27 +/- 0.16 | 0.39 +/- 0.03 | 0.34 +/- 0.27 | 0.26 +/- 0.06 | 0.32 +/- 0.17 | 0.17 +/- 0.04 | 0.27 +/- 0.07 | 0.19 +/- 0.1 | 0.28 +/- 0.13 | 0.27 +/- 0.1 | 0.19 +/- 0.04 | 0.09 +/- 0.12 | 0.26 +/- 0.09 |
| Transcription-repair coupling factor | 0.11 +/- 0.14 | 0.33 +/- 0.01 | 0.29 +/- 0.12 | 0.27 +/- 0.05 | 0.13 +/- 0.11 | 0.22 +/- 0.08 | 0.22 +/- 0.05 | 0.64 +/- 0.94 | 0.26 +/- 0.09 | 0.25 +/- 0.08 | 0.18 +/- 0.12 | 0.23 +/- 0.17 | 0.25 +/- 0.06 |
| Acriflavin resistance protein | 0.32 +/- 0.17 | 0.16 +/- 0 | 0.41 +/- 0.01 | 0.26 +/- 0.1 | 0.16 +/- 0.09 | 0.32 +/- 0.12 | 0.23 +/- 0.06 | 0.23 +/- 0.12 | 0.2 +/- 0.11 | 0.31 +/- 0.08 | 0.5 +/- 0.05 | 0.33 +/- 0.15 | 0.2 +/- 0.1 |
| Alcohol dehydrogenase (EC 1.1.1.1) | 0.24 +/- 0.24 | 0.26 +/- 0.01 | 0.16 +/- 0.08 | 0.25 +/- 0.05 | 0.25 +/- 0.09 | 0.27 +/- 0.04 | 0.27 +/- 0.07 | 0.22 +/- 0.16 | 0.24 +/- 0.09 | 0.26 +/- 0.1 | 0.11 +/- 0.02 | 0.16 +/- 0.11 | 0.31 +/- 0.05 |
| Malonyl CoA-acyl carrier protein transacylase (EC 2.3.1.39) | 0.53 +/- 0.11 | 0.16 +/- 0.01 | 0.28 +/- 0.14 | 0.22 +/- 0.11 | 0.36 +/- 0.17 | 0.15 +/- 0.05 | 0.25 +/- 0.06 | 0.19 +/- 0.12 | 0.2 +/- 0.08 | 0.21 +/- 0.11 | 0.22 +/- 0.04 | 0.2 +/- 0.03 | 0.3 +/- 0.07 |
| Glutamate synthase [NADPH] large chain (EC 1.4.1.13) | 0.45 +/- 0.29 | 0.38 +/- 0.02 | 0.34 +/- 0.15 | 0.26 +/- 0.06 | 0.2 +/- 0.11 | 0.17 +/- 0.05 | 0.27 +/- 0.03 | 0.1 +/- 0.07 | 0.19 +/- 0.14 | 0.26 +/- 0.09 | 0.2 +/- 0.03 | 0.28 +/- 0.16 | 0.24 +/- 0.05 |
| Carbamoyl-phosphate synthase large chain (EC 6.3.5.5) | 0.14 +/- 0.06 | 0.38 +/- 0 | 0.24 +/- 0.09 | 0.25 +/- 0.07 | 0.2 +/- 0.05 | 0.16 +/- 0.03 | 0.24 +/- 0.04 | 0.12 +/- 0.07 | 0.27 +/- 0.28 | 0.31 +/- 0.17 | 0.26 +/- 0.08 | 0.28 +/- 0.16 | 0.17 +/- 0.1 |
| Heat shock protein 60 family chaperone GroEL | 0.2 +/- 0.06 | 0.34 +/- 0 | 0.48 +/- 0.5 | 0.28 +/- 0.08 | 0.2 +/- 0.13 | 0.14 +/- 0.01 | 0.28 +/- 0.04 | 0.08 +/- 0.07 | 0.26 +/- 0.09 | 0.23 +/- 0.11 | 0.48 +/- 0.04 | 0.15 +/- 0.22 | 0.2 +/- 0.08 |
| Thioredoxin reductase (EC 1.8.1.9) | 0.2 +/- 0.19 | 0.24 +/- 0.01 | 0.22 +/- 0.11 | 0.3 +/- 0.08 | 0.18 +/- 0.11 | 0.2 +/- 0.02 | 0.29 +/- 0.09 | 0.16 +/- 0.08 | 0.26 +/- 0.07 | 0.24 +/- 0.1 | 0.11 +/- 0.03 | 0.18 +/- 0.02 | 0.24 +/- 0.03 |
| ClpB protein | 0.24 +/- 0.12 | 0.34 +/- 0.02 | 0.31 +/- 0.1 | 0.28 +/- 0.08 | 0.22 +/- 0.08 | 0.2 +/- 0.03 | 0.26 +/- 0.05 | 0.13 +/- 0.08 | 0.21 +/- 0.09 | 0.27 +/- 0.11 | 0.21 +/- 0.04 | 0.13 +/- 0.19 | 0.17 +/- 0.09 |
| DNA gyrase subunit A (EC 5.99.1.3) | 0.33 +/- 0.11 | 0.3 +/- 0.02 | 0.32 +/- 0.12 | 0.25 +/- 0.05 | 0.21 +/- 0.09 | 0.16 +/- 0.02 | 0.19 +/- 0.04 | 0.16 +/- 0.17 | 0.22 +/- 0.1 | 0.22 +/- 0.12 | 0.37 +/- 0.04 | 0.3 +/- 0.06 | 0.2 +/- 0.04 |
| Multimodular transpeptidase-transglycosylase (EC 2.4.1.129) (EC 3.4.-.-) | 0.14 +/- 0.05 | 0.2 +/- 0 | 0.2 +/- 0.02 | 0.26 +/- 0.12 | 0.23 +/- 0.17 | 0.24 +/- 0.04 | 0.25 +/- 0.05 | 0.17 +/- 0.09 | 0.23 +/- 0.09 | 0.24 +/- 0.15 | 0.09 +/- 0.01 | 0.18 +/- 0.02 | 0.23 +/- 0.08 |
| Alkaline phosphatase (EC 3.1.3.1) | 0.21 +/- 0.04 | 0.07 +/- 0.01 | 0.19 +/- 0.03 | 0.18 +/- 0.06 | 0.16 +/- 0.09 | 0.24 +/- 0.04 | 0.35 +/- 0.05 | 0.19 +/- 0.08 | 0.2 +/- 0.12 | 0.24 +/- 0.14 | 0.4 +/- 0.12 | 0.29 +/- 0.2 | 0.18 +/- 0.08 |
| Asparagine synthetase [glutamine-hydrolyzing] (EC 6.3.5.4) | 0.12 +/- 0.14 | 0.12 +/- 0 | 0.13 +/- 0.02 | 0.15 +/- 0.08 | 0.31 +/- 0.21 | 0.13 +/- 0.03 | 0.17 +/- 0.06 | 0.16 +/- 0.11 | 0.25 +/- 0.12 | 0.29 +/- 0.2 | 0.07 +/- 0 | 0.29 +/- 0.07 | 0.31 +/- 0.17 |
| Translation elongation factor G | 0.24 +/- 0.17 | 0.35 +/- 0.03 | 0.24 +/- 0.11 | 0.25 +/- 0.11 | 0.22 +/- 0.05 | 0.11 +/- 0.04 | 0.22 +/- 0.04 | 0.08 +/- 0.06 | 0.25 +/- 0.21 | 0.24 +/- 0.16 | 0.23 +/- 0.08 | 0.21 +/- 0.29 | 0.2 +/- 0.05 |
| 5-methyltetrahydrofolate--homocysteine methyltransferase (EC 2.1.1.13) | 0.14 +/- 0.12 | 0.3 +/- 0.01 | 0.27 +/- 0.06 | 0.24 +/- 0.1 | 0.19 +/- 0.07 | 0.13 +/- 0.01 | 0.29 +/- 0.05 | 0.11 +/- 0.08 | 0.19 +/- 0.08 | 0.25 +/- 0.11 | 0.16 +/- 0.09 | 0.27 +/- 0.02 | 0.2 +/- 0.12 |
| Threonine dehydrogenase and related Zn-dependent dehydrogenases | 0.13 +/- 0.05 | 0.37 +/- 0.1 | 0.16 +/- 0.02 | 0.27 +/- 0.11 | 0.15 +/- 0.1 | 0.1 +/- 0.04 | 0.3 +/- 0.06 | 0.21 +/- 0.26 | 0.17 +/- 0.1 | 0.25 +/- 0.15 | 0.01 +/- 0.02 | 0.07 +/- 0.09 | 0.29 +/- 0.09 |
| Succinate dehydrogenase flavoprotein subunit (EC 1.3.99.1) | 0.25 +/- 0.02 | 0.29 +/- 0.01 | 0.21 +/- 0.02 | 0.18 +/- 0.08 | 0.19 +/- 0.1 | 0.14 +/- 0.07 | 0.25 +/- 0.04 | 0.15 +/- 0.1 | 0.18 +/- 0.07 | 0.23 +/- 0.12 | 0.24 +/- 0.04 | 0.23 +/- 0.04 | 0.22 +/- 0.08 |
| DNA polymerase I (EC 2.7.7.7) | 0.14 +/- 0.03 | 0.2 +/- 0.01 | 0.4 +/- 0.03 | 0.22 +/- 0.05 | 0.18 +/- 0.13 | 0.21 +/- 0.02 | 0.24 +/- 0.06 | 0.13 +/- 0.11 | 0.19 +/- 0.04 | 0.22 +/- 0.15 | 0.19 +/- 0.01 | 0.2 +/- 0.03 | 0.17 +/- 0.09 |
| Catalase (EC 1.11.1.6) | 0.08 +/- 0.09 | 0.28 +/- 0.03 | 0.15 +/- 0.11 | 0.22 +/- 0.05 | 0.22 +/- 0.14 | 0.15 +/- 0.05 | 0.17 +/- 0.05 | 0.13 +/- 0.09 | 0.28 +/- 0.18 | 0.22 +/- 0.12 | 0.05 +/- 0 | 0.05 +/- 0.08 | 0.21 +/- 0.06 |
| Type I restriction-modification system, restriction subunit R (EC 3.1.21.3) | 0.26 +/- 0.14 | 0.25 +/- 0.01 | 0.25 +/- 0.03 | 0.23 +/- 0.04 | 0.17 +/- 0.04 | 0.31 +/- 0.09 | 0.21 +/- 0.03 | 0.23 +/- 0.1 | 0.15 +/- 0.14 | 0.14 +/- 0.07 | 0.06 +/- 0.03 | 0.35 +/- 0.13 | 0.19 +/- 0.08 |

**References**

Tignat-Perrier, R., Dommergue, A., Thollot, A., Keuschnig, C., Magand, O., Vogel, T. M. and Larose, C.: Global airborne microbial communities controlled by surrounding landscapes and wind conditions, Sci Rep, 9(1), 1–11, doi:10.1038/s41598-019-51073-4, 2019.

---

## Author Response (AR2)

Dear Associate Editor,

We are very pleased that the manuscript has been accepted for publication in Biogeosciences.

We added all the references (for Eggnog mapper, DIAMOND, MEGAN6, SAR NCBI *etc.*) and we modified the sentence that used "preliminary" in the Discussion section.

Best regards,

Romie Tignat-Perrier